# EgoBrain: Synergizing Minds and Eyes For Human Action Understanding

**Nie Lin**[1,*]**, Yansen Wang**[2]**, Dongqi Han**[2]**, Weibang Jiang**[2,*]**, Jingyuan Li**[2,*]**, Ryosuke Furuta**[1]**,
Yoichi Sato**[1,†] **& Dongsheng Li**[2,†]

[1]The University of Tokyo, [2]Microsoft Research Asia

```
{nielin,furuta,ysato}@iis.u-tokyo.ac.jp,
{yansenwang, dongqihan, dongsli}@microsoft.com,
{v-nielin, v-weibangjiang, v-jingyuanli}@microsoft.com,
```

## Abstract

The integration of brain-computer interfaces (BCIs), in particular electroencephalography (EEG), with artificial intelligence (AI) has shown tremendous promise in decoding human cognition and behavior from neural signals. In particular, the rise of multimodal AI models have brought new possibilities that have never been imagined before. Here, we present EgoBrain–the world's first large-scale, temporally aligned multimodal dataset that synchronizes first-person (egocentric) vision and EEG of human brain over extended periods of time, establishing a new paradigm for human-centered behavior analysis. This dataset comprises 61 hours of synchronized 32-channel EEG recordings and first-person video from 40 participants engaged in 29 categories of daily activities. We then developed a muiltimodal learning framework to fuse EEG and vision for action understanding, validated across both cross-subject and cross-environment challenges, achieving an action recognition accuracy of 66.70%. EgoBrain paves the way toward a unified framework for multimodal and egocentric brain–computer interfaces, bridging neural signals and first-person perception. Our dataset and code are publicly available at: `huggingface/ut-vision/EgoBrain` and `github/ut-vision/EgoBrain`.

## 1 Introduction

The explosive growth of artificial intelligence has greatly advanced the field of Brain-computer interfaces (BCI), with massive research efforts to understand brain functions from neural recordings. Among various neural signals, non-invasive systems such as scalp electroencephalograph (EEG) are more scalable, cost-effective, and safer for large-scale adoption (Willett et al., 2021; Anumanchipalli et al., 2019; Sivasakthivel et al., 2025; Metzger et al., 2023; Bai et al., 2023; Li et al., 2025; Lan et al., 2023), thus appealing increasing interest to connect EEG with human perceptions and intentions. Boosted by deep learning techniques, booming breakthroughs have been seen in recent years to decode visual and acoustic stimuli in controlled laboratory settings. For example, recent works achieved accuracies of 15.6% in a 200-way zero-shot task on the EEG-image dataset (Song et al., 2023) and 21.9% in a 9-way task on the EEG-video dataset (Liu et al., 2024b). However, the visual stimuli in existing studies were merely presented on screens and the informative environmental background was ignored. Moreover, the active interactions between the subjects and the environment are less explored due to the passive settings in the experiments.

To better capture real-world human perceptions and actions, we introduce egocentric vision as a complementary modality to EEG. The egocentric vision has emerged as a powerful paradigm for modeling human interactions and perceptual processes in real-world settings, with representative large-scale datasets such as EPIC-KITCHENS(Damen et al., 2022), and Ego4D(Grauman et al., 2022). These datasets primarily capture observable outcomes from a human-like perspective, yielding valuable analysis of human behavior such as action recognition(Zhang et al., 2024; 2025), hand pose estimation(Fan et al., 2024; Lin et al., 2025) and spatial reasoning(Feng et al., 2025).

---

* Work done during an internship at Microsoft Research. † Co-corresponding authors.

However, despite the rapid progress in both EEG decoding and egocentric vision, these two research lines remain fundamentally disconnected. Existing egocentric datasets capture what people do but not what they internally perceive, while traditional EEG studies reveal these internal processes but lack the richness and ecological validity of real-world human–environment interaction. As a result, current benchmarks can only reflect either the external visual outcomes or the internal cognitive responses, but never the interplay between the two. This motivates us to introduce EEG into egocentric vision research, with the goal of filling this critical scientific gap. By pairing real-world egocentric video with simultaneously recorded brain activity, we aim to enable a deeper understanding of how perception and cognition jointly shape human actions.

Interestingly, EEG and egocentric vision provide mutually reinforcing information. While the first-person-view video offers objective information about scenes and actions, the sensorimotor experiences, intentions, and other forms of implicit knowledge remain largely unobservable. The missing pieces can be seamlessly complemented by EEG signals which reveal the latent cognitive signals related to attention, motor planning, decision-making, and intention. Given the complementary nature of egocentric vision and EEG, three fundamental questions arise. First, can their combination lead to a deeper understanding of human behavior? Second, when does this integration outperform unimodal approaches? Third, what technical methodologies can effectively handle the fusion?

To seek the answer and advance human-centric multimodal research, we start from introducing EgoBrain, a large multimodal dataset that synchronously captures EEG and egocentric video from 40 participants engaged in natural daily activities. With a sophisticated design of 29 actions and diverse environmental conditions in the test sets, EgoBrain offers the first benchmark for multimodal action recognition from synchronized EEG and egocentric video. It lays the foundation for multimodal and egocentric brain–computer interfaces that integrate neural activity with first-person vision.

Similar to other multimodal tasks with synchronized timeline, it's crucial to handle the shared temporal structure carefully and fuse information from modalities for downstream prediction. Upon our EgoBrain dataset, we present an adaptive Brain-Time Interval Machine (Brain-TIM) model, inspired from (Chalk et al., 2024) to integrate synchronized visual and EEG signals and capture rich multimodal information for action understanding. Each modality is processed through modality-specific embedding layers and merged to the aggregated global context, while the shared temporal structure is explicitly modeled using the Time Interval MLP (TIM) module. We then conducted experiments with our Brain-TIM to evaluate both the standalone effectiveness of individual modalities and their synergy, and the highlighted results confirmed that the fusion of EEG and vision consistently outperforms unimodal approaches across multiple experiments. Further visualization provide deeper insight into the complementary roles of egocentric vision and EEG signals.

In summary, the main contribution of this paper is threefold:

**1)** We introduce EgoBrain, the first large synchronized EEG dataset designed for egocentric vision research. Featuring data from 40 participants engaged in real-world activities such as tool use and daily tasks (in total 61 hours), this dataset sets a benchmark for cross-modal action understanding and advances the application of BCI technologies in real-life settings.

**2)** To lay the groundwork, we provide standardized preprocessing pipelines for vision-brain synchronization data, along with benchmark evaluations and our proposed Brain-TIM model. These resources ensure experimental reproducibility and offer a unified comparative benchmark for future research based on EgoBrain.

**3)** We conduct ablation studies to assess the individual and combined contributions of different modalities. Our findings offer valuable insights into designing cross-modal learning frameworks for egocentric vision and brain signal integration.

## 2 RELATED WORK

**EEG & Vision Integration:** In recent years, combining electroencephalography (EEG) with visual data has emerged as a central theme in brain–computer interface (BCI) research, elucidating cognitive processes and motor intentions(Mushtaq et al., 2024; Guttmann-Flury et al., 2025; Bertoni et al., 2025; Dreyer et al., 2023; Kaya et al., 2018). EEG's high temporal resolution and portability enable real-time monitoring of brain states, yet most work examines resting-state responses to static visual stimuli, neglecting neural dynamics during natural movement(Yang et al., 2025; Liu

et al., 2025; Ma et al., 2022; 2020; Liu et al., 2024a). A few studies have recorded EEG during active locomotion—for example, assessing cognitive load while walking in a lower-limb exoskeleton(Ortiz et al., 2023)—and virtual-reality tasks like supernumerary thumb control via motor imagery(Alsuradi et al., 2024). However, these efforts target prosthetic control and lack a systematic exploration of real-world, first-person multimodal interactions in unconstrained movement. The recently introduced ToMCAT(Pyarelal et al., 2023) dataset includes diverse tasks such as image rating, virtual table tennis, and Minecraft gameplay. However, unlike EgoBrain, it primarily focuses on screen-based task paradigms and does not capture synchronized egocentric, real-world interactions, leaving the integration of EEG with natural first-person perception largely unexplored.

**Egocentric Vision Datasets:** Recent egocentric video corpora have advanced human–object interaction modeling through varied contexts and annotations(Damen et al., 2022; Grauman et al., 2022; Wang et al., 2023; Darkhalil et al., 2022; Kwon et al., 2021; Liu et al., 2022; Ragusa et al., 2021; Sener et al., 2022; Ohkawa et al., 2023; Zhang et al., 2022; Grauman et al., 2024; Huang et al., 2024). EPIC-KITCHENS(Damen et al., 2022) offers detailed kitchen-activity labels, while Ego4D(Grauman et al., 2022) provides the largest in-the-wild egocentric set for 3D perception and social analysis. HoloAssist(Wang et al., 2023) enables multi-user task completion, and Assemblyseries(Sener et al., 2022; Ohkawa et al., 2023) and H2O(Kwon et al., 2021) cover procedural and two-hand manipulations. More recent datasets like EgoExoLearn(Huang et al., 2024) and Ego-Exo4D(Grauman et al., 2024) deliver asynchronous and dual-perspective recordings of skilled activities. While several egocentric datasets provide multimodal annotations (*e.g.*, audio, IMU, gaze, multi-view footage), none include human-centered internal neural signals such as EEG. As a result, existing resources cannot capture the coupling between brain activity and first-person visual experience, which is the central focus of our work.

Overall, existing research overlooks the synchronization of egocentric visual data and brain activity during dynamic human interactions in daily life.

## 3 EGOBRAIN DATASET

We first introduce the data acquisition setup and collection pipeline, followed by the semantic action design and statistical overview, with the corresponding supplementary details in Appendix B

**Environment & Acquisition System:** Fig.1a illustrates our data capture environment. The setup incorporates lighting and a modular workstation containing objects (books, food, *etc.*) for controlled interactions. The right panel of Fig. 1a illustrates the configuration of our portable recording apparatus. The setup includes a helmet-mounted GoPro HERO12 camera (1080P/30Hz) for capturing high-quality egocentric video and a 32-channel wireless EEG headset (Emotiv FLEX 2 Gel System, 256Hz sampling rate) compliant with the international 10-20 electrode placement standard.

Throughout the session, the subject remains seated to reduce excessive lower-limb movement that may otherwise introduce artifacts into the EEG signals, and the GoPro camera is carefully aligned to the participant's visual horizon to ensure a natural first-person perspective. The subject is asked to conduct some everyday interaction with the objects illustrated in Fig. 1c. Meanwhile, the data acquisition system captures two key modalities: high-fidelity egocentric video recordings and 32-channel EEG signals, with an example shown in Fig. 1b. Recordings were collected from 40 volunteer participants (see Appendix B.1 for subject details).

**Acquisition Pipeline:** Fig. 1d presents a detailed visualization of our standardized action execution pipeline. A session consists of a predefined yet randomly shuffled sequence of 29 actions, with "*Consume*"-related actions repeated for three times (narrated in the next section). At the beginning of each action, a large display screen presents a task prompt (e.g., "*Play Cube*"). The prompt instructs the subject to identify the relevant object placed on the table, initiate the corresponding hand-object interaction. The completion of a task is marked by the subject successfully performing the interaction and manually confirming it via a mouse click. This human-initiated confirmation ensures the intentional execution and completeness of each action, and naturally results in varying action durations across different tasks. After the subject explicitly confirms completion by clicking the "Confirm" button, the system proceeds to the next predefined action. This process continues until the subject has completed the entire set of preprogrammed tasks.

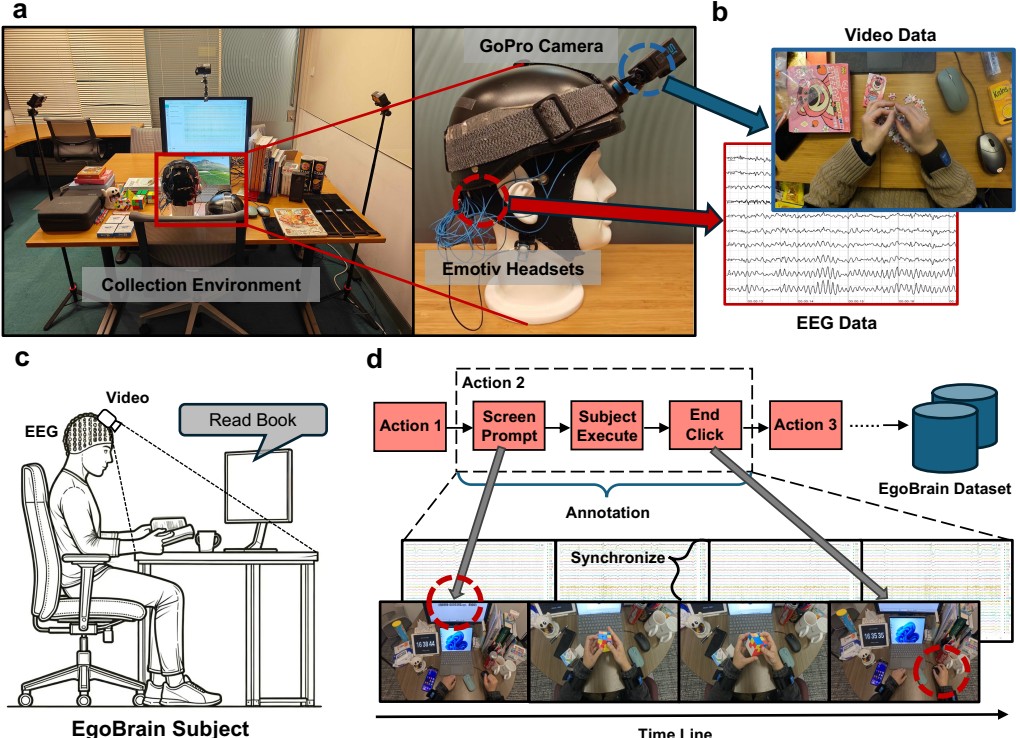

Figure 1: **The EgoBrain dataset and experimental setup. a** (Left) Acoustic isolation chamber with adjustable lighting and modular workstation containing standardized interaction objects. (Right) Portable apparatus configuration showing helmet-mounted GoPro camera and Emotive FLEX 2 Gel EEG headset. **b** High-fidelity egocentric video recording hand-object interactions and 32-channel EEG signals. **c** Subject performing ("*Read book*") action following on-screen textual prompts. **d** From command display ("*Play Cube*") to object interaction and completion confirmation.

Table 1: **Overview of the four high-level activity classes in the EgoBrain dataset.**

| High-Level | Description / Examples |
|---|---|
| **(1) Work** | Operating office software such as Word, PowerPoint, Excel, and Paint, *etc*. |
| **(2) Play** | Engaging in screen-based games, object-based puzzles, and mobile games, *etc*. |
| **(3) Learn** | Performing writing tasks, reading various materials, and drawing activities, *etc*. |
| **(4) Consume** | Eating different types of snacks and drinking multiple beverages, *etc*. |

**Category Design:** The EgoBrain dataset covers a broad spectrum of daily activities, consisting of 29 action classes organized under 10 verbs. We illustrate the design of these semantics in Tab. 1. These four top-level categories offer a coarse yet meaningful structure over the action space, ensuring clear distinctions in cognitive demand, motor behavior, and real-world context. Specifically, "*Work*" includes productivity-oriented computer operations, "*Play*" contains both digital and physical entertainment activities, "*Learn*" captures reading and writing behaviors commonly observed in academic environments, and "*Consume*" reflects everyday eating and drinking actions. To more faithfully capture the diverse cognitive demands and motor behaviors inherent in different forms of "*Play*", we further subdivide the "*Play*" category into three subtypes (see Appendix B.2 for details).

**Evaluation Settings:** To systematically assess model generalization, we define two evaluation settings with increasing difficulty: **Cross-subject-only** and **Cross-subject & Cross-scene**. The Cross-subject-only evaluates subject-level generalization under a shared physical environment and object configuration, whereas the Cross-subject & Cross-scene further introduces a previously unseen environment with different object and background contexts, posing a more challenging cross-domain generalization scenario. Detailed statistics and split configurations are provided in Appendix B.3.

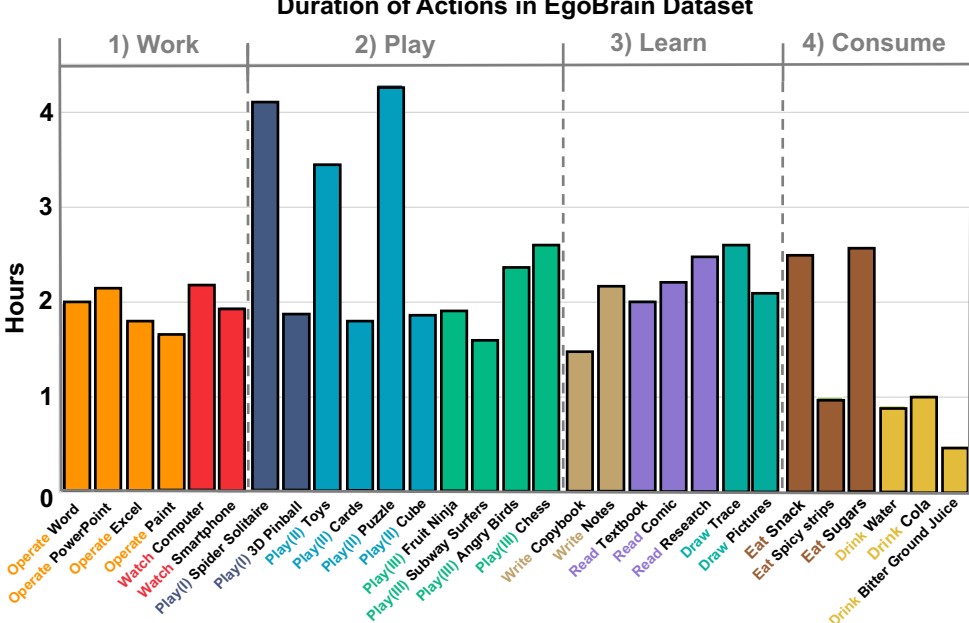

Figure 2: **The EgoBrain statistics.** The total duration per category is presented, highlighting the longest duration (*Play puzzle*: 4.29 hours) and the shortest duration (*Drink Bitter Juice*: 0.49 hours) .

**Dataset Statistics:** These activities span a wide range of temporal scales, with individual task durations ranging from 1,753 seconds (approximately 0.49 hours) to 15,441 seconds (approximately 4.29 hours), reflecting diversity in task complexity. We visualize the cumulative time per activity (in hours) in the Fig. 2. A more comprehensive breakdown, including subject-level distributions and category-wise temporal analysis, is provided in Appendix B.4.

## 4 METHODS

After constructing the EgoBrain dataset, we build an effective framework, namely Brain-TIM, to model these multimodal temporal inputs for action understanding, with additional implementation details provided in Appendix C.

### 4.1 TASK DEFINITION

We consider a time-synchronized pair of raw data: the egocentric video stream and the EEG signal sequence, both sharing a common timeline $\mathcal{T} = [0, T]$. The video stream is represented as $V^{\mathrm{raw}} = \{v_t \in \mathbb{R}^{H \times W \times 3}\}_{t=0}^{T \cdot f^v}$, sampled at a frame rate $f^v$, and the EEG signal as $B^{\mathrm{raw}} = \{b_t \in \mathbb{R}^C\}_{t=0}^{T \cdot f^b}$, recorded at $f^b$ Hz, where $C$ is the number of channels. The target of action recognition task can be formulated as finding the best mapping from input to the action and verb categories $\hat{y} = f_\theta(V, B) \in \{1, \dots, N_c\}^Q$, where $N_c$ equals to 10 for verb classification or 29 for action categories, and $Q$ is the number of consecutive queries which evenly divides the whole time interval $[0, T]$, i.e, the $i$-th query corresponds to the action within time $[(i-1)T/Q, iT/Q]$.

### 4.2 OVERVIEW OF BRAIN-TIM

An overview of Brain-TIM is presented in Fig. 3. We first extract feature representations for each modality using pre-trained backbone networks (Tong et al., 2022; Jiang et al., 2024) into $\phi^v$ and $\phi^b$. These features are then projected into a shared embedding space via modality-specific embedding layers: $g^v$ and $g^b$. The embeddings from different modalities are concatenated to form a unified input sequence for the Transformer module on the right side of the Fig. 3. Eventually, the Transformer

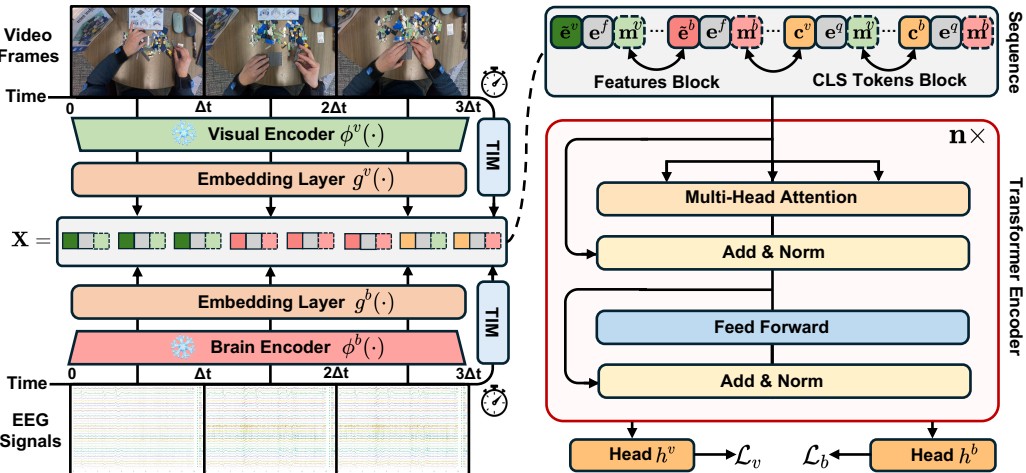

Figure 3: **The overall architecture of Brain-TIM.** The model processes synchronized visual and EEG signals using modality-specific encoders, followed by embedding layers to obtain token sequences. The shared temporal axis is concurrently encoded by the TIM module. A modality-aware `CLS` token is appended to the sequence to capture global semantics. The resulting tokens are fed into a Transformer encoder for downstream action classification.

encoder models the temporal dependencies and cross-modal interactions within the sequence and a linear classifier maps the encoded features to the final action category predictions.

### 4.3 Feature Extraction

Before performing action recognition, we first extract features from all modalities using pre-trained encoders. EEG signals are processed through LaBraM(Jiang et al., 2024), a model pre-trained on 2,500 hours of masked EEG data that generates 2000-dimensional features per channel. Video frames are encoded using VideoMAE(Tong et al., 2022), pre-trained on EPIC-KITCHENS-100(Damen et al., 2022), which outputs 1024-dimensional features per segment. During feature extraction, we adopt the parameter settings for EEG features as suggested in (Jiang et al., 2024), while the parameters for video features follow those introduced in (Chalk et al., 2024).

**Sliding Window Mechanism:** To extract aligned segments, Brain-TIM apply a sliding window mechanism with a duration of $\Delta t$ and step size $\delta t$. Each window contains $N^v = f^v \cdot \Delta t$ video frames and $N^b = f^b \cdot \Delta t$ EEG samples. The raw data are divided into segments aligned with $N = \lfloor \frac{T - \Delta t}{\delta t} \rfloor + 1$, represented as $V = \{\mathbf{v}_i \in \mathbb{R}^{N^v \times H \times W \times 3}\}_{i=1}^N$ and $B = \{\mathbf{b}_i \in \mathbb{R}^{N^b \times C}\}_{i=1}^N$. Below, we detail how to extract feature representations from $V$ and $B$.

The sliding-window mechanism further mitigates the impact of sub-second temporal jitter in a structural manner. In this design, the window stride $\delta t$ is always smaller than the potential misalignment, and adjacent windows exhibit substantial temporal overlap. As a result, each moment in the sequence is covered by multiple windows, creating natural temporal redundancy. Moreover, both modalities are segmented using identical window indices, ensuring that their *relative* temporal structure remains consistent even under slight timestamp shifts.

**Visual Features:** Within each window in $V$, we uniformly down-sample $K$ frames from their corresponding segment, denoted as $\{v_1^i, \ldots, v_K^i\}$. The superscript $i$ refers to the $i$-th window, and $t_i$ is the starting timestamp of the $i$-th window, corresponding to the time interval $[t_i, t_i + \Delta t)$. The timestamps for each sampled frame are denoted as $\{\tau_1^i, ..., \tau_K^i\}$, where $\tau_k^i = t_i + \frac{2k-1}{2K} \cdot \Delta t$, with $k \in \{1, ..., K\}$. This formula ensures that the sampled frames are evenly distributed within the time window and are centered within the window. Each frame is resized to $224 \times 224$ and normalized using ImageNet(Deng et al., 2009) statistics. These $K$ frames are then passed through a frozen, pre-trained visual encoder $\phi^v$ to produce a window-level feature vector $\mathbf{e}^v \in \mathbb{R}^{d^v}$, where $d^v$ represents the feature dimension. Combining the sliding stride $\delta t$, the full video is encoded into a sequence of window-level feature vectors $\mathcal{E}^v = \{\mathbf{e}_1^v, \ldots, \mathbf{e}_N^v\}$, where each $\mathbf{e}_i^v \in \mathbb{R}^{d^v}$.

**Brain Features:** We adopt the same mechanism to extract neural features from $B$. For the $i$-th time window, the EEG signal is denoted as $\mathbf{b}_i$. For the EEG signal within the window, we first apply a band-pass filter with a range of 0.5-50Hz, followed by downsampling to $f^{b'}$ Hz. The input $\mathbf{b}_i$ is fed into a frozen pre-trained encoder $\phi^b$ to obtain its feature representation $\phi^b(\mathbf{b}_i) \in \mathbb{R}^{C \times \Delta t \times d^b}$, where $d^b$ is the EEG feature dimension. The features are aggregated via channel-wise average pooling as $\mathbf{e}^b = \frac{1}{C} \sum_{c=1}^C \phi^b(\mathbf{b}_1, ..., \mathbf{b}_L) \in \mathbb{R}^{\Delta t \times d^b}$, and temporal pooling is applied when necessary. This produces an aligned sequence of window-level features, denoted as $\mathcal{E}^b = \{\mathbf{e}_1^b, \ldots, \mathbf{e}_N^b\}$, where each $\mathbf{e}_i^b \in \mathbb{R}^{d^b}$ and the number of windows $N$ kept consistent with the visual modality.

**Token Preparation:** After the features from both modalities are obtained, the learnable embedding layers $g^v(\cdot)$ and $g^b(\cdot)$ are applied to $\mathcal{E}^v$ and $\mathcal{E}^b$, respectively, to map modality-specific features into a shared $D$-dimensional space. As a result, we obtain the visual feature tokens $\tilde{\mathcal{E}}^v = \{\tilde{\mathbf{e}}_i^v \in \mathbb{R}^D\}_{i=1}^N$ and the EEG feature tokens $\tilde{\mathcal{E}}^b = \{\tilde{\mathbf{e}}_i^b \in \mathbb{R}^D\}_{i=1}^N$ without dimension misalignment for further fusion.

To enable cross-modal interaction and support classification for the $Q$ queries, $2Q$ learnable classification tokens (`cls` tokens) $\{\mathbf{c}_i^v \in \mathbb{R}^D\}_{i=1}^Q$ and $\{\mathbf{c}_i^b \in \mathbb{R}^D\}_{i=1}^Q$ of the same dimension with the feature tokens are introduced for vision and EEG modality, respectively.

### 4.4 SEQUENCE CONCATENATION

**Temporal & Modality-Aware Token:** We enrich all feature tokens $\tilde{\mathbf{e}}_i^v$, $\tilde{\mathbf{e}}_i^b$ and `CLS` tokens $\mathbf{c}_j^v$, $\mathbf{c}_j^b$ with lightweight time-aware embeddings generated by the Time-Interval MLP (TIM), which computes interval-based embeddings such as $\mathbf{e}_i^f$ and $\mathbf{e}_j^q$ from their corresponding temporal ranges.

To distinguish tokens of different modality, we introduced the modality-specific embedding, represented by two learnable vectors, $\mathbf{m}^v \in \mathbb{R}^{2D}$ and $\mathbf{m}^b \in \mathbb{R}^{2D}$, to store shared vision and EEG modality information, respectively. The modality-specific embeddings are directly added to the tokens of the corresponding modality, with detailed formulations provided in Appendix C.1 and Appendix C.2.

**Sequence Concatenation:** The input sequence to the transformer encoder is obtained as follows:

$$\mathbf{X} = \text{Concat}\Big( \underbrace{\{\tilde{\mathbf{e}}_i^v \| \mathbf{e}_i^f + \mathbf{m}^v\}_{i=1}^N}_{\text{visual feature block}}, \underbrace{\{\tilde{\mathbf{e}}_i^b \| \mathbf{e}_i^f + \mathbf{m}^b\}_{i=1}^N}_{\text{brain feature block}}, \underbrace{\{\mathbf{c}_j^v \| \mathbf{e}_j^q + \mathbf{m}^v\}_{j=1}^Q}_{\text{visual CLS token block}}, \underbrace{\{\mathbf{c}_j^b \| \mathbf{e}_j^q + \mathbf{m}^b\}_{j=1}^Q}_{\text{brain CLS token block}} \Big),$$

where each element is constructed by concatenating the original token with its temporal embedding, added to the modality-specific embedding. This final input sequence $\mathbf{X} \in \mathbb{R}^{(2N+2Q) \times 2D}$ is formed by orderly concatenating all processed feature representations and `CLS` tokens. After constructing the final input sequence $\mathbf{X}$, the tokens are processed by a standard Transformer encoder (as detailed in Appendix C.3), followed by a linear classification head (described in Appendix C.4).

This design of Brain-TIM offers three key advantages: 1) it ensures cross-modal time-aware alignment through shared temporal encodings; 2) preserves modality-specific characteristics by utilizing independent modality embeddings; and 3) facilitates cross-modal interaction and query-specific classification by implementing symmetric handling of `CLS` tokens.

## 5 EXPERIMENTS

We rephrase the research questions proposed in the introduction (Sec. 1) here:

**RQ1:** Does a combination of egocentric video and EEG enable a more comprehensive understanding of human behavior?
**RQ2:** Is our proposed method effective for this multimodal action recognition task?
**RQ3:** When does this integration outperform unimodal approaches?

We designed comprehensive experiments to answer these research questions in this section.

### 5.1 ACTION CLASSIFICATION RESULTS ON EGOBRAIN

We evaluate Brain-TIM on test sets of the EgoBrain dataset (Tab. 2) to answer **RQ1**. Note that all experimental results presented in the tables are Mean $\pm$ STD across five different random seeds.

Table 2: **Action recognition results on the EgoBrain test set.** We systematically evaluate unimodal (Brain only, Visual only) and multimodal (Visual+Brain) models under two protocols: **cross-subject only** and **cross-subject & cross-scene**. The table reports the parameter scale (Params) of each model and the mean $\pm$ standard deviation across five random seeds to ensure statistical reliability. The primary evaluation metric is Top-1 accuracy (%), with the best results highlighted in **bold**.

| Protocol | Modality | Encoder | Params | Verb Acc.% | Action Acc.% |
|---|---|---|---|---|---|
| Cross-subject only | Brain only | LaBraM (Jiang et al., 2024) | **5.8M** | $21.53 \pm 0.99$ | $8.44 \pm 2.25$ |
| | Visual only | VideoMAE (Tong et al., 2022) | **305.0M** | $88.95 \pm 0.80$ | $78.44 \pm 0.71$ |
| | Visual + Brain | VideoMAE + LaBraM (Tong et al., 2022; Jiang et al., 2024) | **310.8M** | $\mathbf{90.11 \pm 1.10}$ | $\mathbf{80.16 \pm 1.67}$ |
| Cross-subject & Cross-scene | Brain only | LaBraM (Jiang et al., 2024) | **5.8M** | $19.41 \pm 1.57$ | $9.36 \pm 0.52$ |
| | Visual only | VideoMAE (Tong et al., 2022) | **305.0M** | $81.67 \pm 1.89$ | $63.40 \pm 0.95$ |
| | Visual + Brain | VideoMAE + LaBraM (Tong et al., 2022; Jiang et al., 2024) | **310.8M** | $\mathbf{83.43 \pm 0.41}$ | $\mathbf{66.70 \pm 0.83}$ |

**Unimodal Comparison:** As a well-studied computer vision problem, the visual modality demonstrates significantly strong performance, achieving 88.95% Top-1 accuracy for verb classification and 78.44 % for action classification in the cross-subject setting. Thanks to its superior spatial resolution and contextual richness, egocentric visual input provides fine-grained cues that are critical for distinguishing actions and achieved the performances of 81.67% and 63.40% for verb and action classification even under the challenging cross-subject and cross-scene settings.

As for the EEG modality, the model achieves relatively low yet significantly better performance than chance level for both settings. While EEG data contains certain cognitive information, its relatively low sampling rate and limited feature dimensionality restrict its effectiveness in complex real-world scenarios applied individually. Due to space constraints, we provide the details of the random-baseline results in the Appendix D for further reference.

**Multmodal Comparison:** As shown in Tab. 2, fusing EEG with visual inputs boosts accuracies across both evaluation protocols. Under the cross-subject only setting, the vision-only baseline achieves 78.44% Top-1 accuracy for action classification, while Brain-TIM reaches 80.16%, giving a 1.72% improvement. Specifically, under the most difficult cross-scene setting, the vision-only baseline achieves 63.40% Top-1 accuracy for action classification, while Brain-TIM with both modalities reaches 66.70%, yielding a 3.30% absolute improvement on the 29-class task. This performance gain answers **RQ1** and further highlights the semantic complementarity between the two modalities: while the visual stream captures external manifestations of action, EEG encodes neural signatures of motor intention and implicit knowledge that are not observable from eyes alone.

Importantly, the performance gains of EEG are not attributable to differences in model capacity, but rather to its unique cognitive value. Our EEG encoder is extremely lightweight, with only 5.8M parameters (approximately 1/52 of the visual backbone), yet it still delivers statistically significant improvements. This demonstrates that EEG provides indispensable complementary information in cases of visual ambiguity or occlusion, thereby rendering visual understanding more complete.

## 5.2 ABLATION STUDY

To answer **RQ2** and see whether all the proposed techniques are positively contributing to the decoding task, we removed some components from Brain-TIM and conducted ablation study.

Table 3 presents the performance under three modality settings: Brain Only, Visual Only, and Visual & Brain. For each setting, we evaluate different combinations of three key components: the embedding layer $g^{(v,b)}$, the time interval MLP $I^{(v,b)}$, and the modality embedding $\mathbf{m}^{(v,b)}$.

Overall, each component improves performance in both the pure EEG and multimodal settings. The embedding layer strengthens feature representations. The time-interval MLP helps encode temporal information. The modality embedding preserves modality-specific cues.

Table 3: **Ablation results of Brain-TIM.** These results provide a detailed view of key module contributes to performance under the Brain-Only, Visual-Only, and combined Visual & Brain settings.

| | Brain Only | | | | Visual Only | | | | Visual & Brain | | | | |
|---|---|---|---|---|---|---|---|---|---|---|---|---|---|
| Embedding Layer $g^{(v,b)}$ | ✗ | ✓ | ✗ | ✓ | ✗ | ✓ | ✗ | ✓ | ✗ | ✓ | ✗ | ✗ | ✓ |
| Time Interval MLP $I^{(v,b)}$ | ✗ | ✗ | ✓ | ✓ | ✗ | ✗ | ✓ | ✓ | ✗ | ✗ | ✓ | ✗ | ✓ |
| Modality Embedding $\mathbf{m}^{(v,b)}$ | – | – | – | – | – | – | – | – | ✗ | ✗ | ✗ | ✓ | ✓ |
| Action Acc.% @Top-1 Mean | 7.44 | 7.54 | 6.15 | **9.36** | **64.94** | 64.67 | 64.01 | 63.40 | 65.71 | 65.81 | 66.39 | 66.18 | **66.70** |
| Action Acc.% @Top-1 STD | 0.39 | 0.05 | 0.06 | 0.52 | 3.64 | 1.75 | 6.34 | 0.95 | 0.43 | 2.15 | 0.09 | 2.22 | 0.83 |

Figure 4: **Confusion matrix for verb classification. a** Visual-only. **b** Visual + Brain fusion. Owing to space constraints, verb names are abbreviated; the full abbreviation table is provided in Table 6.

Interestingly, these additional designs reduce performance in the vision-only setting. We suspect that the visual modality alone is already sufficient for accurate predictions, and the additional parameters introduce unnecessary complexity that interferes with the training process.

In an additional ablation, we compare Brain-TIM with a spatial-fusion variant and confirm that temporal fusion achieves better performance on complex action recognition; full details and results are provided in the Appendix E.

## 5.3 DETAILED ANALYSIS

To answer **RQ3**, we further conduct a more detailed analysis of specific categories and representative cases to clarify when the multimodal framework outperforms the unimodal approaches.

**Confusion Matrix of Classification:** We present in Fig. 4 the confusion matrices comparing unimodal and multimodal models for verb classification. Comparing Fig. 4a (Visual-only) and Fig. 4b (Visual + Brain) reveals that EEG integration does not uniformly improve all categories. Notable improvements are seen in "*Play(I)*" (0.46 → 0.64), suggesting EEG complements cognitively demanding actions. The "*Drink*" category benefits from EEG under visual occlusion (0.87 → 0.94).

However, "*Write*" accuracy decreases (0.83 → 0.77), likely due to kinematic redundancy, where EEG introduces noise in cases of clear visual motion patterns. These results indicate EEG's compensatory effect is task-dependent, offering marginal gains when visual cues are strong. Due to space limitations, the complete confusion matrices for all 29 action classes are provided in the Appendix F.

**Benefits of Integrating EEG:** As shown in Fig.5a above, when a subject is drawing in a notebook (*e.g.*, *Santa Claus*), the visual model misclassifies the action as "*Writing*" due to the high visual similarity between the two tasks. However, after incorporating EEG data, the model correctly classifies the action as "*Drawing*". This suggests that EEG signals may capture neural patterns related to task intent and offer additional discriminative cues. It indicates that EEG reflects distinct neural activations associated with visuospatial motor planning, as opposed to language-related tasks—a distinction that has been well documented in prior neuroscience studies(Tang et al., 2024).

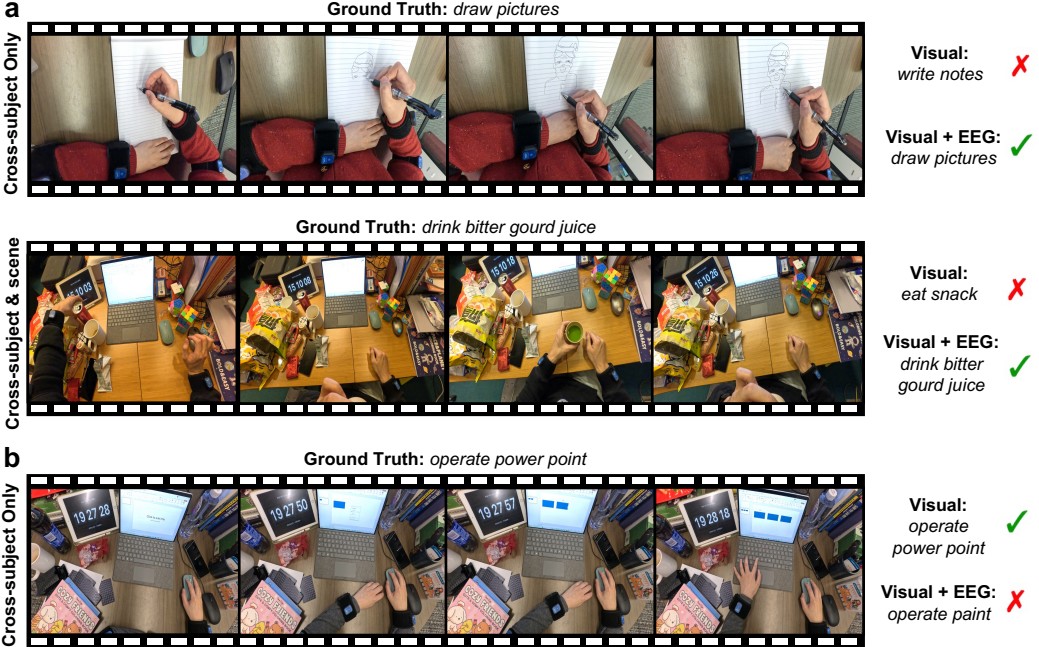

Figure 5: **Success and Failure Cases for Unimodal (Visual) and Multimodal (Visual + EEG) Models.** **(a)** Multimodal model correctly recognizes actions that the visual-only model misses, aided by EEG. **(b)** EEG causes misclassification, possibly due to overlapping cognitive strategies.

Another example in Fig.5a involves the action of "*Drink - bitter gourd juice*". Due to occlusion, the subject's hand and the cup are not visible, and the visual model misclassifies the action as "*Snack*" based on nearby contextual visual cues (*e.g.*, a bag of chips). With EEG integration, the model not only correctly identifies the verb "*Drink*" but also the object "*bitter gourd juice*". This improvement likely stems from EEG's ability to reflect orofacial motor patterns and anticipatory neural activity related to swallowing, which differ from those associated with chewing(Saito et al., 2024). The results underscore EEG's value in disambiguating semantically similar actions when vision is limited.

**Limitations of Integrating EEG:** Despite these advantages, EEG does not always lead to improved recognition. Fig.5b presents failure cases of the multimodal model. The subject is actually operating PowerPoint, but the model incorrectly identifies the task as "*Draw Pictures*". Possible reason is that the subject is creating multiple rectangles, evoking visuomotor activity patterns that resemble those during freehand drawing. Prior studies have shown that such overlapping cognitive strategies lead to similar EEG signatures(Dvorak et al., 2018), making semantic discrimination harder.

## 6 CONCLUSION AND DISCUSSION

We draw on the metaphor that "*the eyes are the windows to the mind*" to argue that egocentric video can illuminate neural states that EEG alone cannot fully capture. Despite its promise, no dataset or systematic study has yet explored EEG–vision synergy in real-world tasks. To address this gap, we construct EgoBrain, the first action understanding dataset that simultaneously captures first-person video and EEG signals, aiming to advance research on vision-brain signal integration. We further develop Brain-TIM as the first multimodal research baseline on EgoBrain. Experimental results show that combining EEG and visual modalities significantly outperforms single-modality approaches, highlighting the potential of multimodal modeling in complex cognitive scenarios.

Our work lays the foundation for applying multimodal brain–computer interfaces to high-level cognitive tasks by introducing a new visuo-neural dataset and an efficient benchmark model. We believe that these contributions will open up new possibilities for brain–vision multimodal learning, and we anticipate future work to actively explore the underlying interaction mechanisms between visual and neural modalities, thereby further inspiring the discovery of new research tasks and directions.

ACKNOWLEDGMENTS

This work was supported by JSPS KAKENHI Grant Number JP24K02956 and JST SPRING, Grant Number JPMJSP2108. This work was also supported in part by Microsoft Research.

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

## A  EEG Preprocessing Pipeline

Following LaBraM(Jiang et al., 2024)'s preprocessing pipeline to ensure compatibility across modalities, we first applied a band-pass filter of 0.1–75 Hz to suppress low-frequency noise, followed by a 50 Hz notch filter to eliminate power-line interference. The signals were then resampled to 200 Hz and normalized by scaling the EEG amplitudes from their raw range ($-0.1$ mV to $0.1$ mV) to approximately $-1$ to 1, with 0.1 mV set as the unit. Unused or noisy channels were removed to further improve signal quality. Finally, raw EEG recordings (e.g., in `.edf` format) were converted into `HDF5` files to facilitate efficient storage and training.

## B  Supplementary Description of the EgoBrain Dataset

To ensure sufficient demographic diversity and enhance the generalizability of multimodal human-centric models, the EgoBrain dataset was collected from a broad and diverse pool of volunteer participants. This section provides an overview of the recruited subjects, the dataset split protocol, finer-grained taxonomy design, and additional visualizations.

### B.1  Subjects

The dataset includes recordings from a total of 40 participants, with a gender ratio of 27 male to 13 female subjects. All subjects were informed of the experimental process and signed informed consent forms before the experiment. This study was approved by the ethical committee of local Institutional Review Board for Human Research Protections.

### B.2  Finer-Grained Taxonomy of Play

As briefly introduced in the main text, the "*Play*" category is further decomposed into three finer-grained subtypes to better reflect differences in cognitive load and behavioral patterns:

- **Play I**: Screen-based games such as "*Spider Solitaire*" and "*3D Pinball*", involving minimal physical movement.
- **Play II**: Object-based activities such as "*Toys*", "*Cards*", "*Puzzles*", and "*Cubes*", requiring moderate motor activity and hand–object interaction.
- **Play III**: Fast-reaction or strategy-oriented mobile games such as "*Fruit Ninja*", "*Subway Surfers*", and "*Chess*", requiring rapid responses or cognitive planning.

Although all three subtypes fall under the same high-level semantic verb "*Play*", they differ substantially in visual appearance, cognitive load, and behavioral patterns. Introducing this finer taxonomy makes the dataset more rigorous. It also helps reduce long-tail effects during data collection and annotation, particularly when modeling multimodal signals involving both vision and EEG.

### B.3  Data Split

We divide the dataset into training, validation, and test sets following the standard data split protocal. To increase the evaluation challenge of the EgoBrain dataset, we design two splits of different difficulty gradient, namely the Cross-subject-only split and the Cross-subject & Cross-scene split. The entire training pipeline strictly follows the standard procedure of selecting the best-performing model on the validation set before conducting the final evaluation on the test set, ensuring fairness and reproducibility in the results.

For the Cross-subject-only setting, we collected 34 different subjects under the same physical environment and object arrangement. These sessions are divided into a train set of 22 subjects (32.96 hours), a validation set of 6 subjects (7.75 hours), and a test set of 6 subjects (9.08 hours).

For the Cross-Subject & Cross-Scene split, we additionally collected 6 new sessions from entirely new subjects in a different environment. These sessions follow the same data collection protocol, but use a completely different object arrangement and take place in a distinct background setting. The train set and validation set consists of 28 subjects (40.71 hours) and 6 subjects (9.08 hours), and

the 6 more sessions (11.28 hours) in the new environment constitutes the test set, assuring that the new environment is never seen during the training and validation stage.

### B.4    TEMPORAL DISTRIBUTION OF ACTIONS

From a temporal standpoint, the longest-duration activities predominantly fall within the "*Play*" and "*Work*" categories. For instance, "*Play Puzzles*" demands sustained attention and intricate hand movements, while "*Watch Computer videos*" or '*Play Games*" can span extended periods. In contrast, actions within the "*Consume*" category are typically brief and episodic. To mitigate under-representation of such short-duration behaviors, we introduced randomized repetition and each "*Consume*"-related action was performed three times during collection.

### B.5    VISUALIZATION ACROSS SUBJECTS

To illustrate the inter-subject diversity inherent in EgoBrain, we present representative initial egocentric video frames from all 40 participants. Fig. 6 and 7 showcase subjects P0001–P0020 and P0021–P0040, respectively. These visualizations highlight substantial cross-subject variability in appearance, posture, and interaction style, reflecting the richness of the collected participant pool and supporting robust generalization in downstream multimodal modeling.

### B.6    VISUALIZATION ACROSS ACTION CATEGORIES

To further demonstrate the breadth of daily activities captured in EgoBrain, we provide visualizations from all 29 annotated action categories. Figures 8–12 present representative egocentric frames across major action types such as operating computers, reading, writing, playing games, and consuming food or beverages. These examples reveal the diversity of visuomotor patterns and contextual scenes within each action class, offering valuable insights into the multimodal dynamics of real-world human behavior.

Overall, the EgoBrain combines demographic diversity, multi-environment robustness, and comprehensive visual recordings, ultimately forming a highly reliable resource for advancing multimodal brain–computer interface research and enabling deeper exploration of real-world human behavior.

## C    SUPPLEMENTARY DESCRIPTION OF THE BRAIN-TIM MODEL

This section provides additional technical details of the Brain-TIM model. Following the same order as described in Sec. 4, we first detail the Time-Interval MLP (TIM) for temporal encoding, then the modality-specific embeddings, followed by the Transformer encoder architecture, and finally the classification heads and training objective.

### C.1    TIME-AWARE TOKEN EMBEDDING

To incorporate the time-specific information to the tokens, we explicitly add it by introducing the Time-Interval MLP (TIM), $I(\cdot) : \mathbb{R}^2 \to \mathbb{R}^D$, consisting of three linear layers with ReLU activations and LayerNorm operation. This TIM module takes the start and end time $(t_s, t_e)$ of the corresponding interval as input and generate the temporal embedding which can be further appended to the specific token as a time-aware token embedding.

For the feature tokens $\tilde{\mathbf{e}}_i^v$ and $\tilde{\mathbf{e}}_i^b$, the time interval is determined by the $i$-th window $[t_i, t_i + \Delta t)$, also named feature time in (Chalk et al., 2024). The temporal embedding is thus calculated as $\mathbf{e}_i^f = I(t_i, t_i + \Delta t) \in \mathbb{R}^D$. As for the CLS tokens $\mathbf{c}_j^v$ and $\mathbf{c}_j^b$, the time interval corresponds to the $j$-th query $[(j-1)T/Q, jT/Q]$, also known as query time. And the temporal embedding is similarly obtained as $\mathbf{e}_j^q = I((j-1)T/Q, jT/Q) \in \mathbb{R}^D$.

### C.2    MODALITY-SPECIFIC EMBEDDING

To distinguish tokens of different modality, we further introduced the modality-specific embedding, represented by two learnable vectors, $\mathbf{m}^v \in \mathbb{R}^{2D}$ and $\mathbf{m}^b \in \mathbb{R}^{2D}$, to store shared vision-modality

and EEG-modality information, respectively. The modality-specific embeddings are directly added to the tokens of corresponding modality.

## C.3 TRANSFORMER ENCODER

As shown in the Transformer encoder on the right side of Fig. 3, the input sequence $\mathbf{X}$ is processed by a stack of Transformer encoder layers. Each layer consists of a self-attention mechanism followed by a feedforward network, following the architecture proposed in (Vaswani et al., 2017). The self-attention mechanism enables the model to capture long-range dependencies across different positions in the sequence, while the feedforward network applies non-linear transformations to the input. Each layer uses residual connections and layer normalization to facilitate gradient flow, with the output passed to the next layer to refine the input sequence $\mathbf{X}$ representations.

## C.4 LINEAR CLASSIFICATION

Following the Transformer Encoder, as shown in the bottom right of Fig. 3, we extract the modality-specific and query-specific CLS tokens from the output sequence. These tokens are fed into their respective classification heads, $h^v$ and $h^b$, which consist of linear layers followed by softmax to produce class probabilities. The two modality branches are trained and evaluated independently, without merging their predicted probabilities into a single unified prediction.

The model is supervised using modality-specific cross-entropy losses, denoted as $\mathcal{L}_v$ and $\mathcal{L}_b$ for the visual and EEG branches, respectively. The total loss function $\mathcal{L}$ is defined as the sum of the visual modality loss $\mathcal{L}_v$ and a weighted term for the EEG modality loss $\mathcal{L}_b$, scaled by a hyperparameter $\lambda$:

$$\mathcal{L} = \mathcal{L}_v + \lambda \cdot \mathcal{L}_b.$$

It is worth highlighting that the high-level semantic structure (*e.g.*, "*Work*", "*Play*", "*Learn*", "*Consume*") is intentionally not used during training. The three-tier hierarchy ("high-level category $\rightarrow$ verb $\rightarrow$ action") was introduced primarily to conceptually abstract and organize human daily activities. As such, the highest level serves merely as a semantic scaffold for dataset users and is not suitable to function as a supervisory signal for model optimization.

In contrast, our classification design explicitly incorporates the two semantic levels used during training—verbs and actions. These labels are fully leveraged in our training pipeline: the embeddings produced by the Transformer encoder are passed into two separate classification heads—one for verbs and one for actions—each equipped with its own loss function. This design enables the model to learn behavior representations at multiple levels of semantic granularity, while avoiding the potential biases that may arise from enforcing overly coarse, concept-driven category supervision.

# D    ADDITIONAL EXPERIMENTAL RESULTS

**Random-baseline Result:**    To provide a clearer sense of the lower bound performance on the EgoBrain benchmark, we report several standard random baselines for both verb and action classification. These baselines help contextualize the difficulty of the task and offer reference points for interpreting the multimodal models.

We evaluate three commonly used forms of random performance: (1) uniform chance level, (2) prior-based random sampling that follows the empirical class-frequency distribution, and (3) the majority-class baseline. The results are summarized in Table 4.

Table 4: **Random baselines for EgoBrain verb and action classification.**

| Metric | Verb Classes | Action Classes |
|---|---|---|
| Chance Level | 10.00% | 3.45% |
| Prior-based Random | 11.77% | 4.02% |
| Majority Class | **18.59%** | **7.02%** |

Among these, the majority-class classifier yields the strongest random performance, with 7.02% accuracy for action classification. To contextualize the Brain-only model under the challenging Cross-Subject & Cross-Scene protocol, we compare its performance against this upper-bound random baseline. The Brain-only model achieves $9.36 \pm 0.52$, corresponding to a 33.3% relative improvement over the majority-class baseline.

These baselines demonstrate that, despite the substantial inter-subject variability and the inherent difficulty of EEG-based prediction in real-world scenarios, the Brain-only model consistently surpasses all random and prior-driven strategies. At the same time, the gap between these baselines and the Brain-only performance indicates meaningful room for future progress on EgoBrain, inviting further exploration from the research community.

**Robustness Comparison:** Interestingly, the Brain-only model shows slightly higher action accuracy under the cross-subject & cross-scene protocol. This arises because EEG captures head-centered neural dynamics that remain largely invariant across environments, so increasing the training set from 22 to 28 subjects directly enhances its discriminability. In contrast, VideoMAE depends on visual appearance and motion cues that shift significantly with background, lighting, and object changes, leading to strong degradation under scene variation. Consequently, EEG benefits from richer cross-subject diversity, whereas VideoMAE suffers from cross-scene domain shift.

# E    TEMPORAL FUSION *vs.* SPATIAL FUSION

Our Brain-TIM adopts a temporal fusion strategy to integrate visual and EEG modalities when constructing the multimodal input sequence $\mathbf{X}$. In contrast, a simpler alternative way is to concatenate modality features along the spatial dimension while keeping the fusion module active but effectively removing one modality's tokens. We refer to this baseline as *spatial fusion*. Under this formulation, the input $\mathbf{X}$ is formed as:

$$\mathbf{X} = \text{Concat}\big( \underbrace{\{\tilde{\mathbf{e}}_i^v \| \tilde{\mathbf{e}}_i^b \| \mathbf{e}_i^f\}_{i=1}^N}_{\text{feature block}}, \underbrace{\{\mathbf{c}_j^v \| \mathbf{c}_j^b \| \mathbf{e}_j^q\}_{j=1}^Q}_{\text{CLS token block}} \big) \in \mathbb{R}^{(N+Q)\times 3D}.$$

The key difference between the two fusion paradigms lies in how modality-specific structure is preserved. Brain-TIM uses dedicated modality embeddings $\mathbf{E}_V$ and $\mathbf{E}_E$ to explicitly encode visual and EEG token identities before temporal fusion. In contrast, the spatial-fusion baseline directly concatenates the visual and EEG representations:

$$\mathbf{X}\text{fusion} = \text{Concat}(\mathbf{X}_V, , \mathbf{X}_E) \in \mathbb{R}^{B\times (V+E)},$$

and treats the concatenated tensor as a single fused modality without distinguishing token types.

Table 5: Comparison of temporal fusion (Brain-TIM) and spatial fusion. Temporal fusion clearly improves performance for the more challenging action recognition task.

| Setting | Verb Acc. | Action Acc. |
|---|---|---|
| Vision Only | $81.67 \pm 1.89$ | $63.40 \pm 0.95$ |
| Visual & Brain (Spatial fusion) | $83.74 \pm 0.62$ | $64.81 \pm 1.04$ |
| Visual & Brain (Brain-TIM) | $83.43 \pm 0.41$ | $\mathbf{66.70 \pm 0.83}$ |

To ensure consistency with our main experimental protocol, all results were averaged across five different random seeds. The comparison between the two fusion strategies is summarized in Table 5. While both methods achieve comparable performance on verb classification, Brain-TIM with temporal fusion substantially outperforms spatial fusion on the more challenging 29-way action recognition task. This demonstrates that although simple feature concatenation can suffice for coarse semantic discrimination, explicitly modeling temporal dependencies across time steps leads to more robust and discriminative multimodal representations—particularly beneficial for fine-grained, real-world action recognition.

# F    CONFUSION MATRIX OF ACTION

Table 6: **Verb-level and action-level abbreviations used in the EgoBrain dataset.** Each verb is assigned a two-letter code ("Verb code"). Each fine-grained action is assigned a two-letter "Action code", and the final label is formed as `Verb_Action`.

| Verb | Verb code | Action | Abbreviation |
|------|-----------|--------|--------------|
| Operate | OP | Operate Word | `OP_WD` |
| | | Operate PowerPoint | `OP_PP` |
| | | Operate Excel | `OP_EX` |
| | | Operate Paint | `OP_PT` |
| Watch | WA | Watch Computer | `WA_CP` |
| | | Watch Smartphone | `WA_SP` |
| Play (I) | P1 | Play(I) Spider Solitaire | `P1_SS` |
| | | Play(I) 3D Pinball | `P1_PB` |
| Play (II) | P2 | Play(II) Toys | `P2_TY` |
| | | Play(II) Cards | `P2_CD` |
| | | Play(II) Puzzle | `P2_PZ` |
| | | Play(II) Cube | `P2_CB` |
| Play (III) | P3 | Play(III) Fruit Ninja | `P3_FN` |
| | | Play(III) Subway Surfers | `P3_SS` |
| | | Play(III) Angry Birds | `P3_AB` |
| | | Play(III) Chess | `P3_CH` |
| Write | WR | Write Copybook | `WR_CB` |
| | | Write Notes | `WR_NT` |
| Read | RD | Read Textbook | `RD_TB` |
| | | Read Comic | `RD_CM` |
| | | Read Research Paper | `RD_RP` |
| Draw | DR | Draw Trace | `DR_TR` |
| | | Draw Pictures | `DR_PC` |
| Eat | ET | Eat Snack | `ET_SN` |
| | | Eat Spicy strips | `ET_SS` |
| | | Eat Sugars | `ET_SG` |
| Drink | DK | Drink Water | `DK_WT` |
| | | Drink Cola | `DK_CL` |
| | | Drink Bitter gourd juice | `DK_BG` |

To maintain clarity in the main paper, we omit the full 29-way action confusion matrices due to their size and the limited space available. These matrices, however, offer useful insights into fine-grained model behavior and characteristic error patterns. To present them compactly and coherently in the supplementary material, we adopt the verb–action abbreviation scheme in Tab. 6, where each action is encoded using a concise two-level code. This scheme preserves semantic interpretability while enabling a cleaner and readable visualization of the dense $29 \times 29$ matrices. In Fig. 13 and Fig. 14, we provide the complete confusion matrices for the visual-only and visual-brain models, offering a more comprehensive view of their respective error distributions and class-separation behavior.

Taken together, the two confusion matrices offer a detailed view of how visual and multimodal models behave in fine-grained egocentric action recognition. The Visual-only model tends to struggle with actions that share similar hand trajectories, object appearances, or desktop-level contexts, which leads to noticeable clusters of confusion in several verb groups. When EEG is incorporated, the Visual+Brain model shows a general trend toward stronger diagonal concentration and reduced cross-class ambiguity, suggesting that neural signals may provide complementary cues that help differentiate visually similar actions. Overall, these results indicate that multimodal integration can improve recognition in scenarios where vision alone is limited or ambiguous.

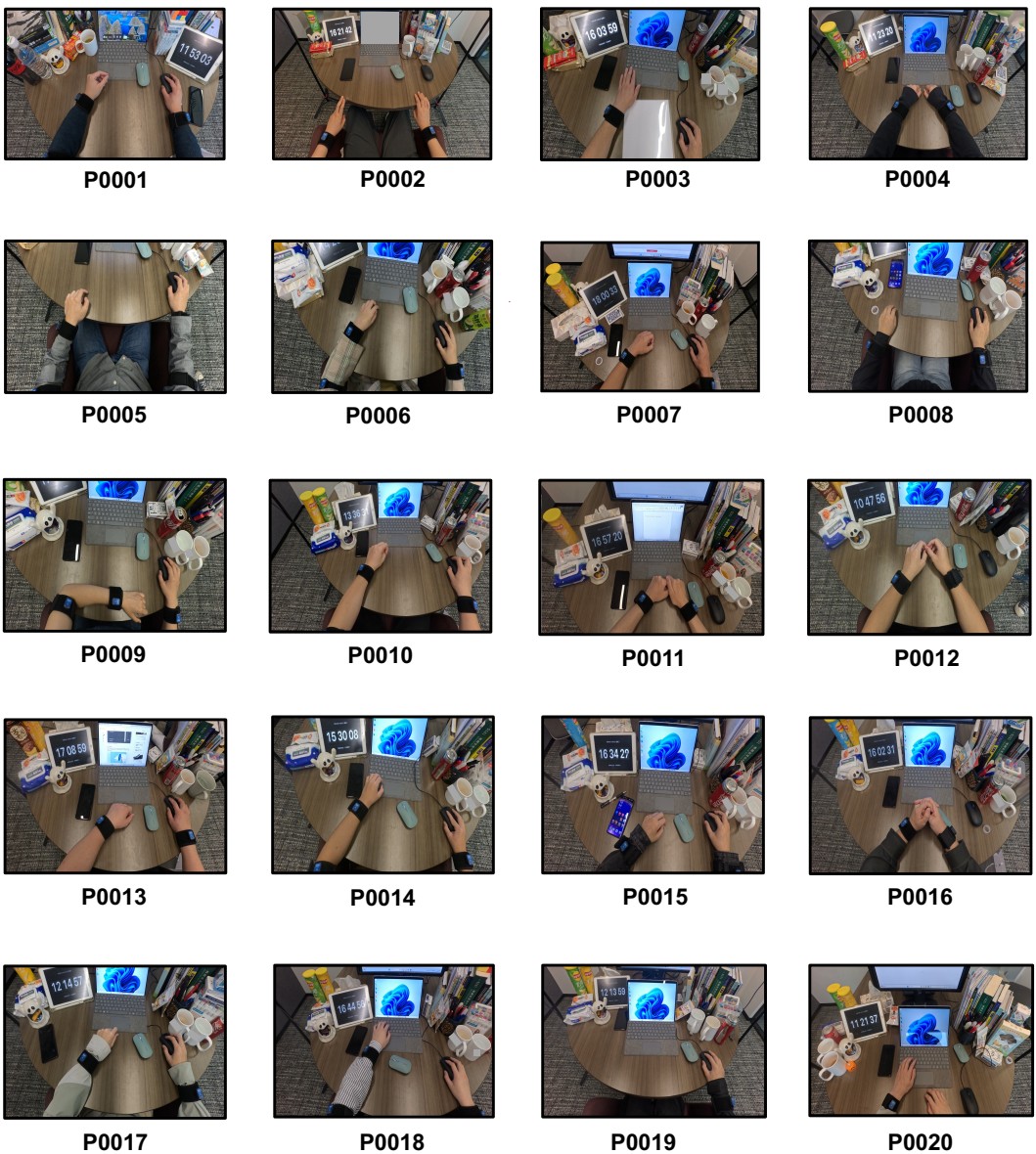

Figure 6: **We provide visualizations of initial video frames from participants: P0001 to P0020.**

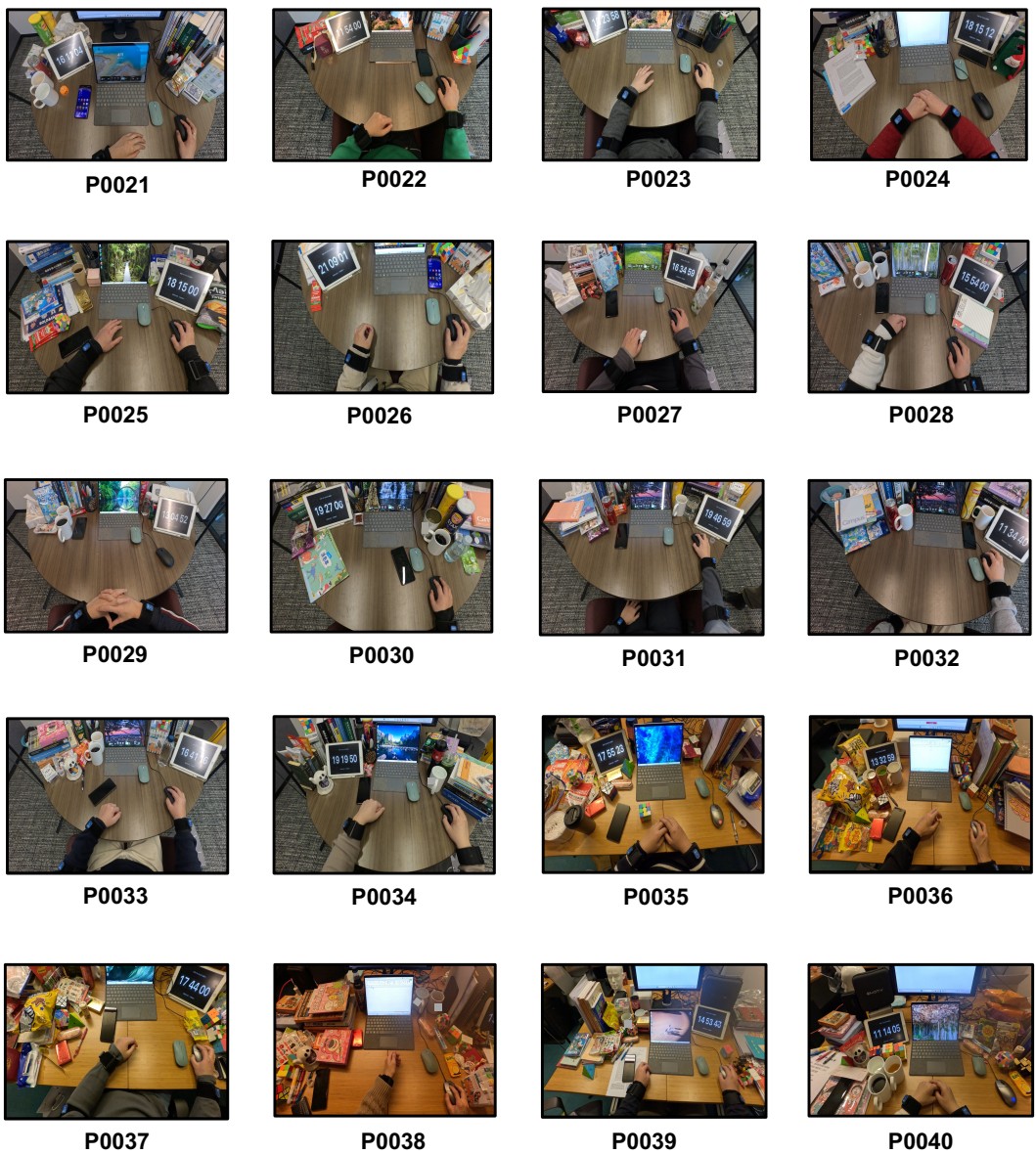

Figure 7: **We provide visualizations of initial video frames from participants: P0021 to P0040.**

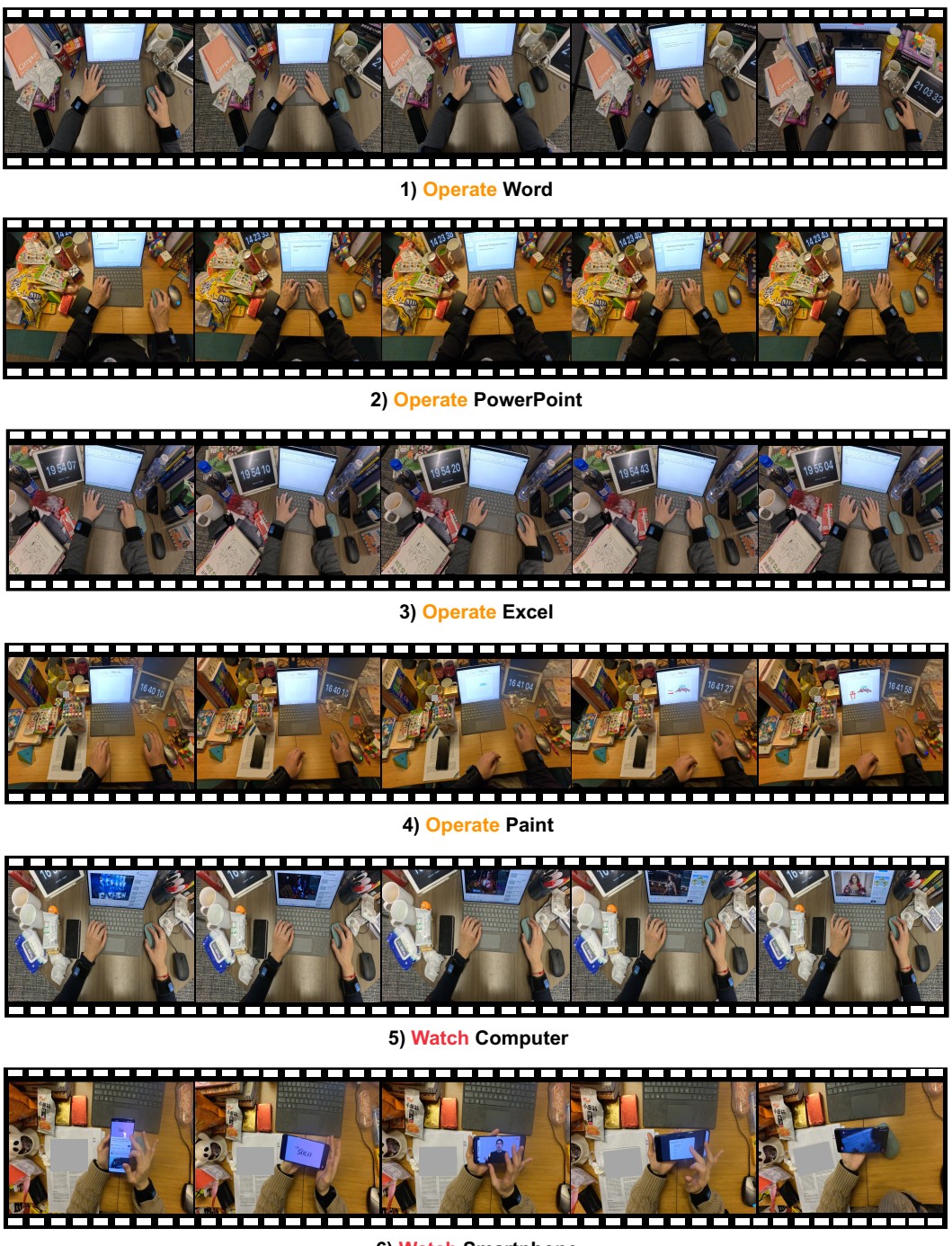

Figure 8: **Visualization of selected action categories including "*Operate*" and "*Watch*".** The egocentric perspective in each sequence offers intuitive insight into the subject's ongoing motor behavior.

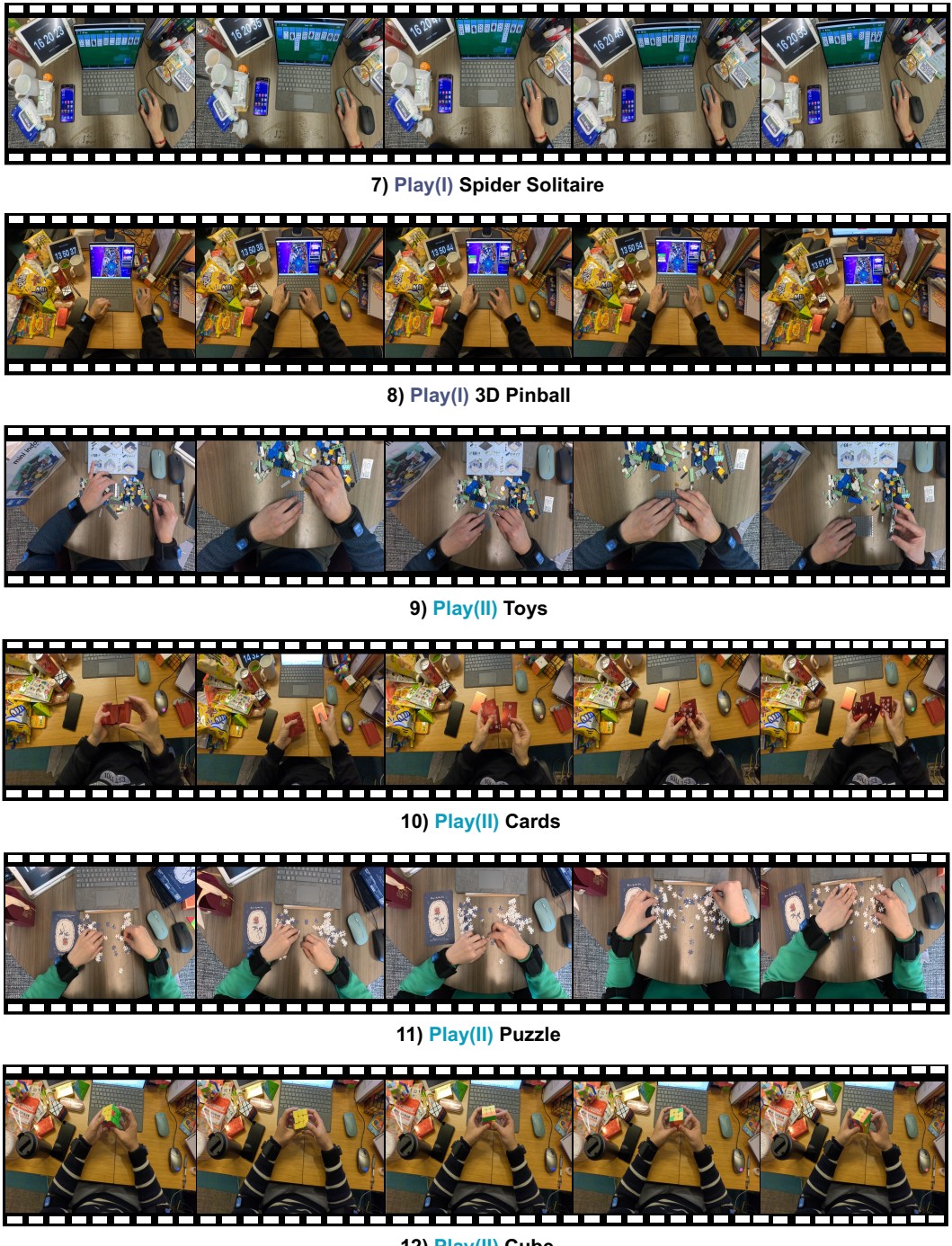

Figure 9: **Visualization of selected action categories including "*Play(I)*" and "*Play(II)*".** The egocentric perspective in each sequence offers intuitive insight into the subject's ongoing motor behavior.

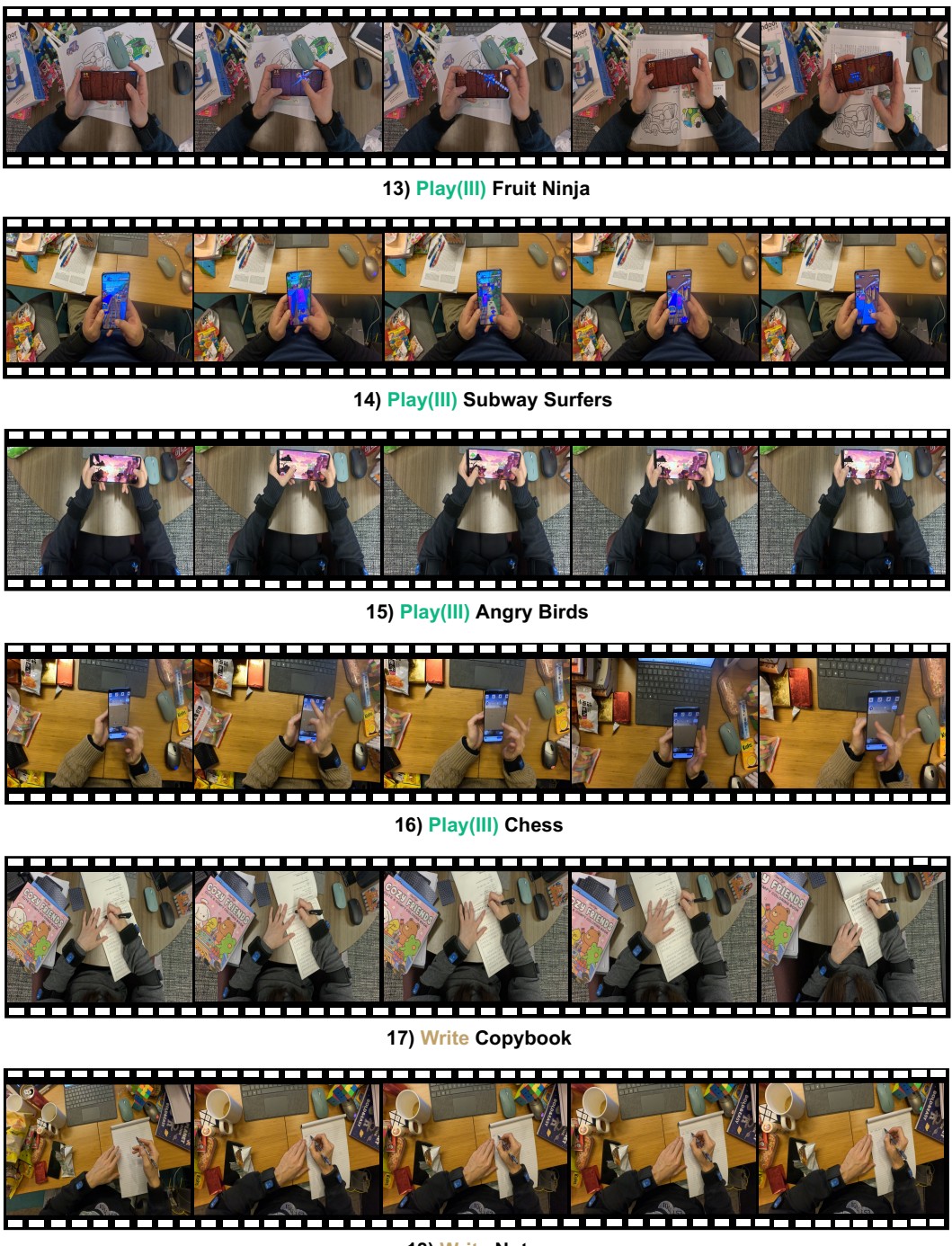

**13) Play(III) Fruit Ninja**

**14) Play(III) Subway Surfers**

**15) Play(III) Angry Birds**

**16) Play(III) Chess**

**17) Write Copybook**

**18) Write Notes**

Figure 10: **Visualization of selected action categories including "*Play(III)*" and "*Write*".** The egocentric perspective in each sequence offers intuitive insight into the subject's ongoing motor behavior.

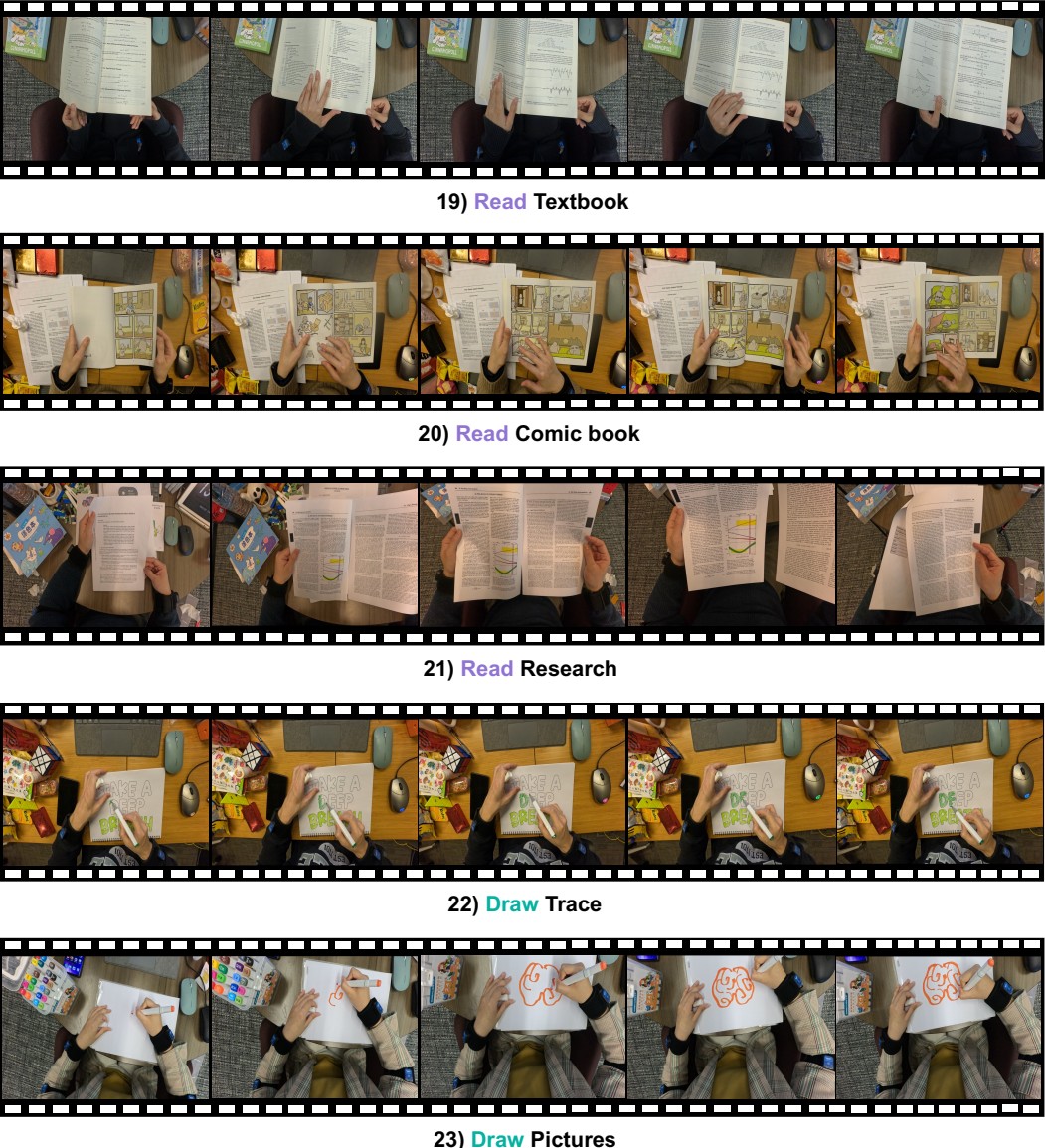

**19) Read Textbook**

**20) Read Comic book**

**21) Read Research**

**22) Draw Trace**

**23) Draw Pictures**

Figure 11: **Visualization of selected action categories including "*Read*" and "*Draw*".** The ego-centric perspective in each sequence offers intuitive insight into the subject's ongoing motor behavior.

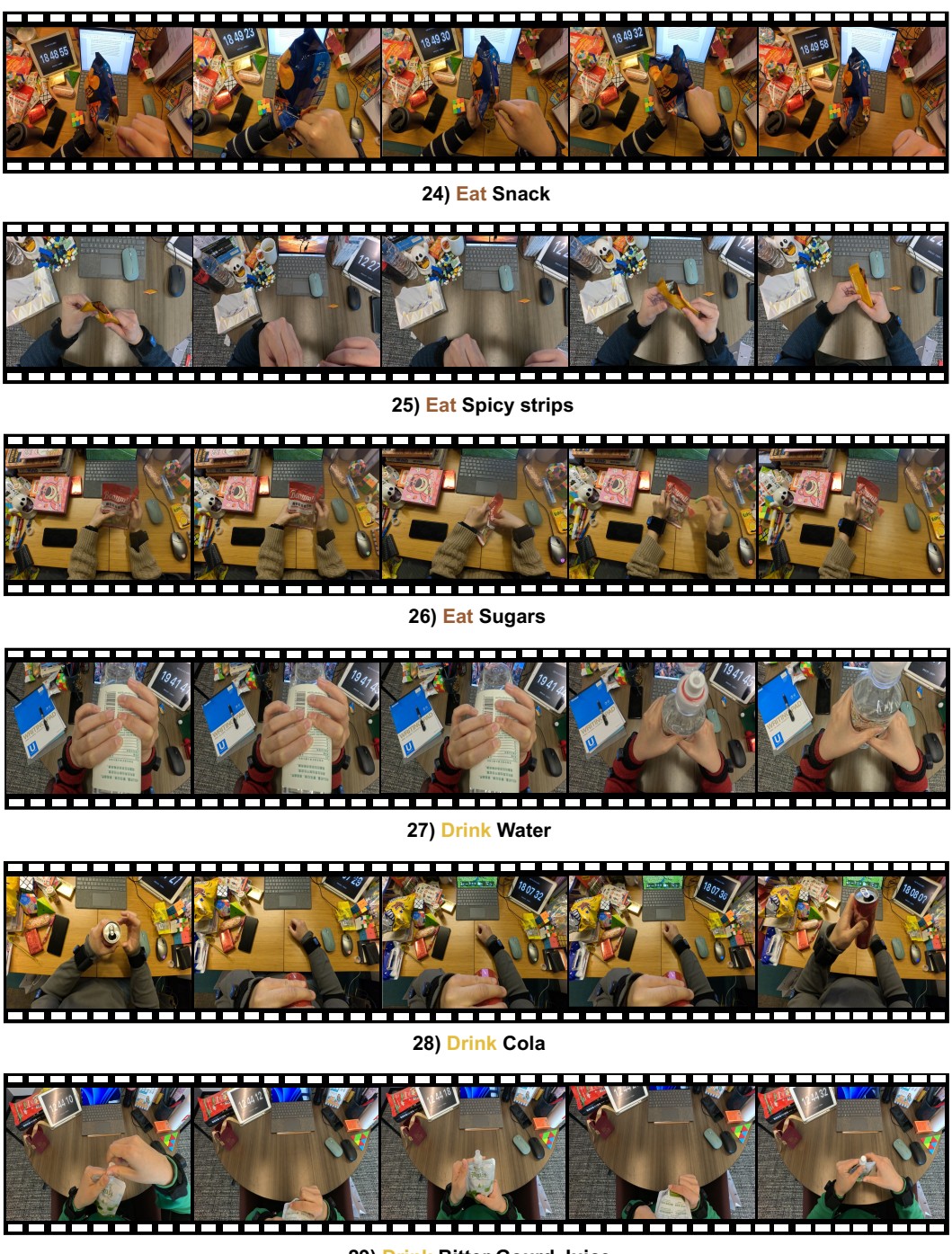

**24) Eat Snack**

**25) Eat Spicy strips**

**26) Eat Sugars**

**27) Drink Water**

**28) Drink Cola**

**29) Drink Bitter Gourd Juice**

Figure 12: **Visualization of selected action categories including "*Eat*" and "*Drink*".** The egocentric perspective in each sequence offers intuitive insight into the subject's ongoing motor behavior.

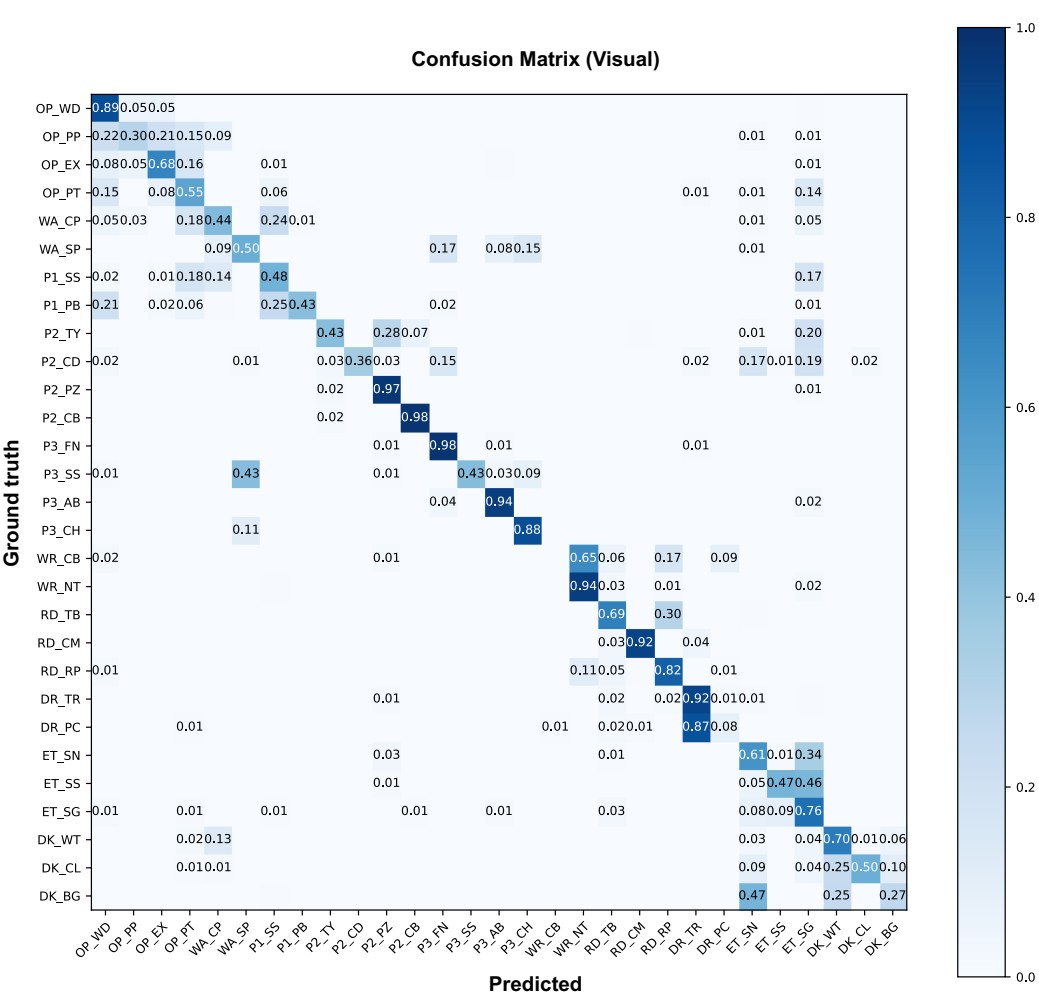

Figure 13: **29-way action confusion matrix of the Visual-only model.**

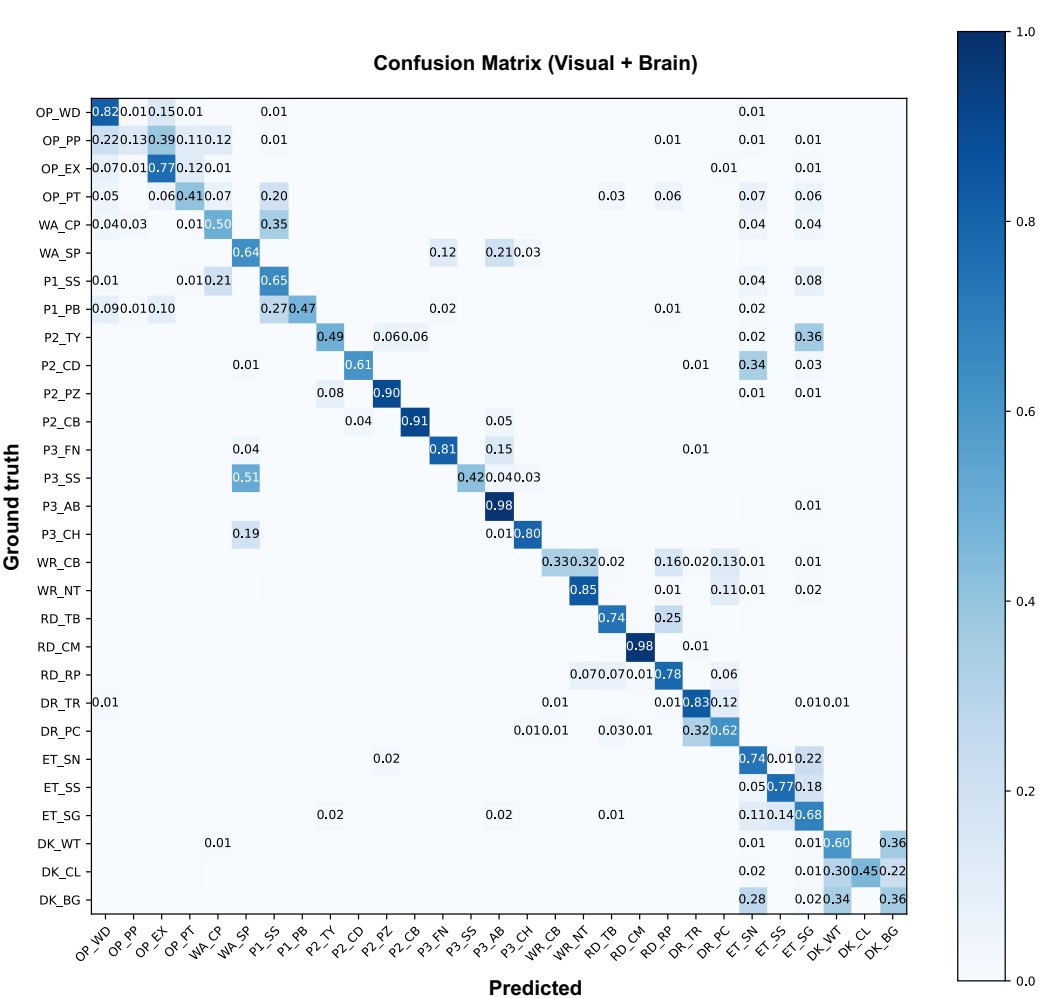

Figure 14: **29-way action confusion matrix of the Visual+Brain (multimodal) model.**

