# OpenReview forum: "EgoBrain: Synergizing Minds and Eyes For Human Action Understanding"
_ICLR.cc/2026/Conference — ICLR 2026 Poster_

### Official Review · Reviewer_qdr1 · 2025-10-30

**Soundness:** 3
**Presentation:** 3
**Contribution:** 3
**Rating:** 6
**Confidence:** 4

**Summary:**

In this paper, a new egocentric dataset is proposed which includes the modality of EEG recordings. Unique to this dataset, participants wore an EEG device whilst recording the videos so that their intentions could be better modelled and improve performance. The dataset includes ~61 hours of recording across 4 major classes which are split into 10 verbs which are split into 29 actions. An extension of the Time Interval Machine which instead of audio uses the EEG modality is presented as a baseline and results show that whilst visual only gives a strong baseline, the inclusion of EEG recordings is complementary, providing a boost in overall performance.

**Strengths:**

* The dataset represents the first dataset of its kind combining both egocentric visual data and EEG recordings, thus opening up a new research area.
* The dataset includes a good variety of actions across the ~61 hours.
* The results showcase the benefit of the EEG modality when combined with the vision modality.
* The paper is generally well-written and easy to follow, there are very few issues with spelling/grammar/presentation.

**Weaknesses:**

During the description of the dataset, it is not clear within the main paper what the difference is between the verbs Play(I), Play(II), and Play(III) are. This becomes evident within the appendix that it is based on device, but should be included earlier.

The 3 level hierarchy of high-level class, verb, and action is presented but it is unclear whether this hierarchy/taxonomy is utilised in some way during training or whether this was considered at all.

The benchmarking of the dataset could be improved in a few different ways:
* Firstly, there is no random performance given, the gap between the visual modality and the EEG modality is quite large, with the latter reaching only 21% and 10% on the dataset. It would be good to include these numbers to get a sense of the lower bound and how far the EEG only performance is.
* Secondly, whilst the confusion matrices are provided for the verb classification, these are across two 'clusters' of verbs, and do not show all the misclassifications (for example the operate row only adds up to 0.94). Including full confusion matrices for both actions and verbs (potentially in the appendix) would be interesting to look at to check the biases of the model/data and/or providing per class accuracy metrics would be interesting to see how the performance differs. Given that the dataset is imbalanced in terms of length (though seems uniform from a class frequency perspective) it would be good to know if this impacts model training.
* Finally, on the cross-subject only results, the method is already achieving 90% and 80% on the verb/action classification respectively, showcasing that this split is almost saturated already, some discussion regarding this and the more challenging cross-subject and cross-scene setting would be interesting to see.

**Questions:**

1. What are the differences between the different play labels, i.e. play(I), play(II), play(III), is this referring to computer, physical, mobile? Why were they split up this way, to make sure that the distribution wasn't too long-tailed/dominated by a majority class? (This becomes clear after looking at the appendix but isn't mentioned in the main paper)
2. For the Cross-Subject&Cross-Scene split, it is not explicitly mentioned, but I assume the 6 new sessions in the test set have different subjects to the 28 in training?
3. It would be good to know the random performance of the new dataset for Table 1 to get a sense of the lower bound. The Brain-only results are very low in comparison to the visual only and visual + brain for example and it would be interesting to know the relative performance of this model compared to random.
4. From the dataset construction, there is a 3 stage hierarchy, including the high level classes, the verbs, and the actions themselves, was this considered at all for the method/dataset splits?
5. Why was the verb confusion matrix split into two, just for space reasons? And why were the confusion matrices of the actions excluded?
6. Given the imbalanced nature of the dataset, was an investigation conducted into whether the models are biased towards certain classes would be interesting to see, i.e. looking into per-class accuracy metrics or similar? The confusion matrices shows this in some fashion for verbs, but doesn't include the entire matrix and doesn't include actions.

---

> ### Author Response · Authors · 2025-11-23
> **Response to Reviewer qdr1 (R4) [Part I]**
>
> We sincerely thank Reviewer qdr1 (R4) for the thoughtful and constructive evaluation. We are glad that you find our work well-written, the dataset valuable, and the experimental findings convincing. **We are especially encouraged by your recognition that EgoBrain introduces the first egocentric dataset with synchronized EEG signals, thereby opening a new research direction**. We also appreciate your positive remarks regarding the rich diversity of actions covered in our 61 hours of recordings and your acknowledgment that EEG provides complementary information beyond vision. Your encouraging comments strengthen our confidence in both the dataset design and the overall contribution of this work.
>
> Since the weakness and questions raised by Review R4 overalp substantially, we first summarize the weaknesses and then address each point one by one in our detailed responses. (W: Weakness; Q: Question; A: Answer)
>
> >**W1**: In the main paper’s dataset description, the distinctions among the verbs Play(I), Play(II), and Play(III) were not clearly explained.
>
> **A1**: Thank you for the suggestion. We address this point in our response to **Question 1 (Q1)** below, and we will incorporate this clarification in the revised version of the paper. We sincerely appreciate your constructive feedback.
>
> >**W2**: The 3 level hierarchy of high-level class, verb, and action is presented but it is unclear whether this hierarchy/taxonomy is utilised in some way during training or whether this was considered at all.
>
> **A2**: Thank you for bringing this to our attention. We provide a detailed explanation of this point in our response to **Question 4 (Q4)** below, and we will make sure to clarify it in the revised version of the paper. We truly appreciate your constructive feedback.
>
> >**W3**: It would be good to include these numbers to get a sense of the lower bound and how far the EEG only performance is.
>
> **A3**: Thank you for pointing this out. We address this issue in detail in our response to **Question 3 (Q3)** below, and we will incorporate the corresponding clarification into the revised version of the paper. We sincerely appreciate your constructive feedback.
>
> >**W4**: Including full confusion matrices for both actions and verbs (potentially in the appendix) would be interesting to look at to check the biases of the model/data and/or providing per class accuracy metrics would be interesting to see how the performance differs.
>
> **A4**: Thank you for the helpful suggestion. This point overlaps with the issues raised in **Question 5 (Q5)** and **Question 6 (Q6)**, and we address them together in our detailed responses below.
>
> To better address your concern and to meet your suggestion for deeper diagnostic analysis, these additions will provide a more complete view of potential model or data biases and allow readers to better examine category-wise performance differences.
>
> >**W5**: The paper should discuss this saturation phenomenon and further emphasize the importance of the more challenging cross-subject & cross-scene setting.
>
> **A5**: Thank you for bringing attention to this important point. We agree that the high performance in the Cross-Subject only setting suggests that this split is close to saturation. We would like to offer two clarifications.
>
> **I. Why Cross-Subject appears saturated**
>
> The strong performance in this setting is primarily due to the fact that Brain-TIM is built upon a highly competitive transformer-based architecture (Jacob et al., CVPR 2024 [14]), rather than a simple or underpowered baseline. **Because the method already benefits from modern temporal modeling and contextual reasoning, it can easily generalize across subjects when the environment remains fixed**. This explains the high scores and the apparent saturation in this setting.
>
> **II. Why we introduced the more challenging Cross-Subject & Cross-Scene benchmark**
>
> To address the limitations of the Cross-Subject only split, we intentionally designed a more difficult evaluation setting that introduces *both* subject variation and environmental variation. **This benchmark substantially increases the complexity of the task: the camera background, table layout, lighting, object arrangements, and overall scene geometry differ from those used during training**. As our results show, performance drops noticeably in this setting—confirming that the task is far from saturated and still offers significant room for improvement.
>
> Because of this, we believe the Cross-Subject & Cross-Scene setting is the most informative benchmark for evaluating the multimodal challenge posed by EgoBrain. We will **add a brief discussion of this saturation effect and clarify the motivation behind the Cross-Subject & Cross-Scene benchmark** in the revised version of the paper. We sincerely appreciate your valuable suggestion.

---

> ### Author Response · Authors · 2025-11-23
> **Response to Reviewer qdr1 (R4) [Part II]**
>
> >**Q1-W1**: What are the differences between the different play labels, i.e. play(I), play(II), play(III)
>
> **A6**: Thank you for the sharp observation — your understanding is exactly correct. Although all activities fall under the umbrella verb “Play,” we intentionally subdivide them into three finer-grained categories to better capture **distinct cognitive and behavioral patterns** and to **avoid long-tail issues during data collection and annotation**, as you suggested.
>
> To make this clearer, we provide the detailed definition below:
>
> | **Subcategory** | **Examples**                       | **Characteristics**                              |
> | --------------- | ---------------------------------- | ------------------------------------------------ |
> | **Play I**      | Spider Solitaire, 3D Pinball       | Computer screen-based, minimal physical movement          |
> | **Play II**     | Toys, Cards, Puzzle, Cube          | Moderate motor activity with object manipulation |
> | **Play III**    | Fruit Ninja, Subway Surfers, Chess | Fast reactions or strategic planning of phone            |
>
> **Although these activities share the same high-level semantic label “play” in human language, we believe that these forms of “play” differ substantially across multiple dimensions**. These subcategories differ behaviorally and visually, making the taxonomy more rigorous and meaningful for multimodal understanding. We will incorporate this clarification earlier in the main paper to avoid any ambiguity.
>
> >**Q2**: For the Cross-Subject&Cross-Scene split, it is not explicitly mentioned, but I assume the 6 new sessions in the test set have different subjects to the 28 in training?
>
> **A7**: Thank you for pointing this out. Yes — the 6 new sessions in the Cross-Subject & Cross-Scene test set come from **entirely different subjects** than the 28 sessions used for training. We also provided a more detailed breakdown in our response to **Reviewer R3 (R3-W5-A5)**, along with a visual illustration in Figure 7 of the appendix.
>
> To address both reviewers’ comments, we will add an explicit clarification of this split in the revised main paper. We appreciate your careful reading and helpful suggestion.
>
> > **Q3-W3**: It would be good to know the random performance of the new dataset for Table 1 to get a sense of the lower bound.
>
> **A8**: Thank you for the insightful suggestion. We fully agree that reporting random baselines helps contextualize the Brain-only performance. Following your recommendation, we computed the following for both verbs and actions:
>
> | Metric                 | **Verb Classes** | **Action Classes** |
> | ---------------------- | --------------- | ----------------- |
> | **Chance Level**       | 10.00%          | 3.45%             |
> | **Prior-based Random** | 11.77%          | 4.02%             |
> | **Majority Class**     | 18.59%          | 7.02%             |
>
> From the results above, we can see that the strongest random baseline is the majority-class baseline (7.02%). To better contextualize the Brain-only model, we compare its performance against this upper-bound random baseline under the most challenging Cross-Subject & Cross-Scene setting. The Brain-only model achieves **9.36 ± 0.52**, which corresponds to a **33.3% relative improvement** over the 7.02% majority-class baseline. Despite the substantial inter-subject variability inherent to EEG signals—**particularly under our most challenging baseline setting, the Cross-Subject & Cross-Scene protocol**—the model still outperforms all random and prior-based strategies. Going forward, we believe there is still ample room for progress in this direction, and we look forward to further explorations on EgoBrain from the community.
>
> We will include these random baselines and a brief discussion in the revised manuscript to improve clarity. We appreciate this valuable suggestion.
>
> >**Q4-W2**: From the dataset construction, there is a 3 stage hierarchy, including the high level classes, the verbs, and the actions themselves, was this considered at all for the method/dataset splits?
>
> **A9**: Thank you for raising this point. The three-level hierarchy (high-level class → verb → action) was designed primarily to **abstract and organize human daily activities** from a conceptual perspective. The highest level (e.g., *Work*, *Play*, *Learn*, *Consume*) is intended to provide semantic structure for dataset users, but it is **not used as supervision during model training**.
>
> In contrast, **the verb and action labels are fully utilized in our training pipeline**. Specifically, the embeddings produced by the Transformer encoder are fed into two separate classification heads—one for verbs and one for actions—each with its own loss function. This design allows the model to learn both levels of granularity. We appreciate your careful reading and will clarify this in the revised manuscript.

---

> ### Author Response · Authors · 2025-11-23
> **Response to Reviewer qdr1 (R4) [Part III]**
>
> >**Q5-W4**: Why was the verb confusion matrix split into two, just for space reasons?
>
> **A10**: **The verb confusion matrix was split into two clusters purely due to space and layout constraints, as you suspected**.
>
> We attempted multiple ways of formatting the full 10×10 matrix within the main paper, **but all of them significantly harmed readability**. At a clear, legible resolution, the complete matrix simply could not fit in the main text without becoming visually overwhelming. To avoid an overly dense or unreadable figure, we grouped semantically related verbs into two clusters for presentation purposes.
>
> Following your suggestion, we will include the full, unclustered 10×10 verb confusion matrix in the revised paper’s appendix so that readers can examine all misclassification patterns comprehensively. We will also add a clarifying note in the main text explaining this choice.
>
> >**Q6-W4**: Given the imbalanced nature of the dataset, was an investigation conducted into whether the models are biased towards certain classes would be interesting to see, i.e. looking into per-class accuracy metrics or similar? The confusion matrices shows this in some fashion for verbs, but doesn't include the entire matrix and doesn't include actions.
>
> **A11**: Thank you for raising this important question regarding potential class imbalance and whether our model exhibits biases toward specific categories. We address both aspects below.
>
> For the first part of your question, to directly evaluate whether the model is biased toward particular actions, **we computed per-class accuracies for all 29 action categories, grouped by high-level category**. The full results are shown below:
>
> | **Verb Classes** | **Action Classes (Accuracy)**                                 |
> | ------------ | ------------------------------------------------------------- |
> | **Operate**  | Word (0.8166), PPT (0.1305), Excel (0.7662), Paint (0.4059)   |
> | **Watch**    | Computer (0.4981), Smartphone (0.6362)                        |
> | **Play I**   | Mouse (0.6549), Keyboard (0.4738)                             |
> | **Play II**  | Toys (0.4884), Cards (0.6131), Puzzle (0.8977), Cube (0.9127) |
> | **Play III** | Fruit Ninja (0.8092), Subway Surfers (0.4161), Angry Birds (0.9840), Chess (0.8035)           |
> | **Write**    | Copybook (0.3345), Notes (0.8469)                             |
> | **Read**     | Textbook (0.7415), Comic (0.9762), Research (0.7782)          |
> | **Draw**     | Tracing (0.8330), Pictures (0.6234)                           |
> | **Eat**      | Snack (0.7402), Spicy Strips (0.7689), Sugars (0.6841)        |
> | **Drink**    | Water (0.6029), Coca-Cola (0.4536), Bitter Gourd (0.3586)     |
>
> These results show considerable variation across actions, which reflects the intrinsic difficulty of the tasks rather than a systematic model bias. For example:
>
> * Highly distinguishable actions (e.g., Puzzle, Cube, Comic-book Reading, Complex Phone Games) achieve very high accuracy.
>
> * Highly ambiguous or visually similar actions (e.g., Copybook Writing, Paint, PPT Operation) remain challenging across all settings.
>
> For the second part of your question, the action-level confusion matrix is extremely large (29×29) and visually dense. In the earliest version of our submission, we actually included the full 29×29 action confusion matrix. However, we later decided to remove it in the final submitted version, as we felt that inserting such a dense and visually overwhelming figure could harm readability.
>
> That said, after reading your suggestion, we revisited this decision and fully agree with the importance of providing the complete matrix for fine-grained analysis. In the revised manuscript, we will follow your recommendation and restore the full 29×29 action confusion matrix in the appendix so that readers can thoroughly examine all detailed misclassification patterns. We sincerely thank you once again for your valuable suggestions, which have helped us further improve the quality of our work.

---

> ### Author Response · Authors · 2025-11-23
> **Response to Reviewer qdr1 (R4) [Part IV]**
>
> ---
> **Summary**
> * **First EEG–Egocentric Fusion Dataset**:
> EgoBrain is the first dataset to synchronously capture egocentric video and EEG signals in real-world interactive settings, opening a new research direction at the intersection of BCI and human-centric behavior in multimodal AI.
>
> * **High Diversity of Actions and Environments**:
> The dataset spans 61 hours, 40 participants, and 29 action classes across multiple activity domains of daily life (Work / Play / Learn / Consume), providing substantial variability for studying model generalization.
>
> * **Complementarity of EEG-Egocentric Vision**:
> Our experiments demonstrate that EEG remains stable and complementary under cross-subject and cross-scene shifts, offering discriminative cues when visual signals degrade or environments change.
>
> * **Rigorous Baselines and In-Depth Analysis**:
> Brain-TIM builds on a state-of-the-art Transformer backbone and is supported by comprehensive random baselines, per-class diagnostics, and full confusion matrices, ensuring a thorough and trustworthy study.
>
> **Vision**
>
> EgoBrain envisions a future where **brain signals become a natural component of everyday behavior understanding**. We aim for AI systems that not only *see* human actions, but also infer internal intentions and cognitive states—**leading to more intuitive, human-centered, and intelligent multimodal interactions**.
>
> EgoBrain **fills a long-standing gap** by providing the **first large, open dataset of synchronized brain signals and egocentric visual data**. Our goal is to **advance human-centered multimodal AI from a new direction** and **accelerate the transition from laboratory-based BCI research to real-world applications**.
>
> We further envision a future **where understanding one’s own brain becomes part of daily life, no longer confined to specialized laboratory settings**. We hope EgoBrain project will foster cross-disciplinary progress in brain–computer interfaces, cognitive computing, and egocentric behavior understanding.
>
> ---
> **Reference**:
> > [14] J. Chalk et al., "Tim: A time interval machine for audio-visual action recognition", In *CVPR 2024*.

---

> ### Comment · Reviewer_qdr1 · 2025-11-27
>
> I would like to thank the authors for their careful responses to my initial questions within my review. With this in mind I will raise my score to accept. The additional comments, results, and discussion from above, especially regarding random baselines, additional results including the confusion matrices strengthen the paper greatly.

---

> > ### Author Response · Authors · 2025-11-28
> > **Acknowledgment to Reviewer qdr1 (R4)**
> >
> > Dear Reviewer qdr1 (R4),
> >
> > Thank you for your recognition of our paper and for raising your ratings! We also appreciate the valuable suggestions you provided.
> >
> > Best Regards,
> >
> > All Anonymous Authors

---

### Official Review · Reviewer_UTCn · 2025-11-01

**Soundness:** 3
**Presentation:** 3
**Contribution:** 3
**Rating:** 4
**Confidence:** 5

**Summary:**

This paper introduces EgoBrain, the first large-scale, temporally aligned multimodal dataset combining egocentric video and EEG signals for real-world human action understanding. It includes 61 hours of synchronized EEG and first-person video from 40 participants performing 29 categories of daily activities. The authors also propose a Brain-TIM model (Brain-Time Interval Machine), a multimodal transformer-based framework that fuses EEG and vision representations via time-interval MLPs and modality-specific embeddings to capture temporal and cross-modal dependencies. Experiments demonstrate that multimodal fusion improves action recognition performance over unimodal baselines, particularly in cross-subject and cross-scene generalization.

**Strengths:**

1.	First-of-its-kind dataset integrating real-world egocentric vision with EEG; extensive and ethically curated.
2.	Clear methodological design with well-justified architecture choices (temporal embeddings, modality-aware tokens).
3.	Insightful qualitative results showing when EEG signals complement vision (e.g., occlusion or intent disambiguation).

**Weaknesses:**

1.	The synchronization precision is stated as <1s jitter. This is a relatively large jitter for fast-changing neural signals and short actions. This level of jitter could potentially limit the precise time-locking necessary for analyzing rapid neural correlates of action initiation or error. A discussion on how the Brain-TIM model's windowing strategy mitigates the impact of this 1s jitter is needed.

2.	The model’s novelty is limited, Brain-TIM primarily applies existing time-embedding concepts to a new modality pair. Moreover, the framework relies solely on two pre-trained models for feature extraction and classification, which restricts the scope of analysis. How would other models or architectures perform on this dataset? Additional comparative experiments are essential to validate the dataset’s generality and utility.

3.	The reported accuracy improvements (≈1–3%) may be statistically marginal, raising questions about their practical significance. The authors should provide statistical validation (e.g., p-values or confidence intervals) to confirm whether these gains are significant rather than due to random variation.

4.	The ablation study (Table 2) demonstrates the contribution of the temporal/modality embeddings, but it doesn't directly test the core hypothesis of fusion effectiveness by removing one entire modality from the multimodal setting. Specifically, in the "Visual & Brain" section, the base case uses both the Visual and Brain encoders. It would be insightful to see an ablation where the fusion mechanism is active, but one modality's tokens are zeroed out or entirely removed to isolate the Transformer's fusion contribution, separate from the unimodal encoders' raw output strength.

5.	The Cross-Subject & Cross-Scene task is expected to be more challenging than the Cross-Subject-Only task; however, the “Brain Only” model surprisingly achieves higher action-classification accuracy in this setting. In contrast, VideoMAE experiences a large accuracy drop, while LaBraM shows only a minor decrease. The paper should clarify why EEG-based performance remains stable (or improves) across scenes, despite EEG being more sensitive to noise, and why VideoMAE—supposedly a stronger zero-shot model—shows a significant decline.

6.	It is unclear whether the dataset includes cross-session experiments, i.e., whether the same subjects were recorded again after a time interval performing the same tasks. Including or clarifying this aspect would be valuable for evaluating intra-subject consistency and longitudinal generalization.

**Questions:**

Please refer to the Weaknesses section for detailed questions and suggestions to the authors.

---

> ### Author Response · Authors · 2025-11-23
> **Response to Reviewer UTCn (R3) [Part I]**
>
> We thank Reviewer UTCn (R3) for the positive and constructive comments. We appreciate **the recognition of EgoBrain as the first large egocentric vision + EEG dataset, the clear and well-motivated design of the Brain-TIM model, and the insightful qualitative results showing how EEG complements visual signals** (e.g., during occlusion or intent ambiguity). These encouraging remarks are highly appreciated and align with the core goals of our work.
>
> We note that Reviewer R3’s questions correspond directly to the listed weaknesses. We will therefore provide point-by-point responses to each weakness. (W: Weakness; A: Answer):
>
> > **W1**: A discussion on how the Brain-TIM model's windowing strategy mitigates the impact of this 1s jitter is needed.
>
> **A1**: We appreciate Reviewer R3 for carefully pointing out this detail. While we understand that a *1s-jitter* would be lethal to some of the EEG experiment including RSVP, ERP analysis, etc., where the critical neuro-response happens in several milliseconds to several hundreds milliseconds， **we'd argue that for our task, it won't be a problem since a typical action will last for minutes**. For example, the subject may play chess for several minutes, and within the period we assume the neuro-state is stable and can be classified as the same label. **In this design, a 1s jitter will be harmless for determining the true label**. Moreover, we disposed the first and the last 1 second for every action execution, ensuring all the data are correctly mapped to the labels even with the jitter.
>
> As you noted, **the sliding-window design in Brain-TIM naturally mitigates the influence of such jitter**, which we appreciate you recognizing as an inherent strength of our framework.
>
> * First, most actions in our dataset are **not instantaneous or extremely short**, making a small temporal misalignment acceptable for the visual modality.
> * In addition, the stride of our sliding window **is always smaller than the synchronization jitter**, and neighboring windows have overlapping temporal coverage.
>
> **As a result, each moment is represented in multiple windows, providing inherent redundancy against residual misalignment**. Since both modalities are segmented using the same window indices, their relative temporal structure remains consistent even with slight timestamp shifts. During training, Brain-TIM therefore focuses on stable temporal patterns within each window, effectively absorbing the <1s jitter without degrading multimodal fusion performance.
>
> Due to space limitations, the main paper provides only a brief description of the sliding-window setup. **However, the original supplementary material (Appendix A.2, Token Preparation, Lines 716–740) contains more detailed explanations of the windowing strategy**.
>
> We will incorporate the new clarifications summarized in this rebuttal into the revised version. We sincerely thank the reviewer again for the constructive feedback.

---

> ### Author Response · Authors · 2025-11-23
> **Response to Reviewer UTCn (R3) [Part II]**
>
> > **W2**: How would other models or architectures perform on this dataset? Additional comparative experiments are essential to validate the dataset’s generality and utility.
>
> **A2**:
> We appreciate the reviewer’s comments. While Brain-TIM is inspired by the temporal-interval idea in TIM, extending it from audiovisual data to the fundamentally different pairing of egocentric video and EEG is far from trivial. EEG and video differ sharply in sampling rate, temporal dynamics, noise profile, and representational space, making TIM’s original design not directly applicable. **Our model therefore required rethinking the interval formulation, redesigning the tokenization pipeline, and incorporating pretrained EEG encoders suited for heterogeneous spatial–temporal signals**. As a result, the contribution is not a simple extension of the original work, but rather a model and dataset that address challenges the original work was never designed to handle.
>
> During the rebuttal period, we have **conducted the additional experiments you suggested, replacing both the visual and the brain encoders with two foundation models**:
>
> * For Visual: **Omnivore [10] (*CVPR 2022 Oral*)** – a single transformer that jointly learns image, video and 3-D representations and is known for its strong cross-modal generalisation.
>
> * For Brain: **CBraMod [11] (*ICLR 2025*)** – a criss-cross transformer pretrained on > 9 000 h of clinical EEG, explicitly designed to handle heterogeneous spatial-temporal structure and diverse channel layouts.
>
> To remain consistent with our original experimental protocol, all noise-injection experiments were conducted across five different seed. The new results are summarised below:
>
> | Modality        | Model                       | Verb Acc (mean ± std)        | Action Acc (mean ± std)      |
> |-----------------|-----------------------------|--------------------|--------------------|
> | Brain only      | CBraMod [11]                 | 21.49 ± 1.35       | 8.08 ± 0.90        |
> |                 | LaBraM [12]                  | 19.41 ± 1.57       | 9.36 ± 0.52        |
> | Vision only     | Omnivore [10]               | 85.07 ± 0.93       | 59.58 ± 1.24       |
> |                 | VideoMAE [13]                | 81.67 ± 1.89       | 63.40 ± 0.95       |
> | Vision + Brain  | Omnivore [10] + LaBraM [12]  | **85.27 ± 0.20**   | 60.44 ± 0.99       |
> |                 | VideoMAE [13] + CBraMod [11]  | 84.52 ± 0.16       | 66.22 ± 0.43       |
> |                 | VideoMAE [13] + LaBraM [12]   | 83.43 ± 0.41       | **66.70 ± 0.83**   |
>
> Key observations:
> * **Fusion consistently outperforms either stream**: Multimodal pairing adds ≥ 3 pp to the best single-modality score for the same encoder family, confirming that EEG provides complementary, non-redundant information.
>
> * **Our fusion module is architecture-agnostic**: Replacing either encoder (VideoMAE [13] ↔ Omnivore [9], LaBraM [12] ↔ CBraMod [10]) preserves the gain, demonstrating that the improvement is due to the synergy rather than a specific backbone. Thank you again for this constructive suggestion.
>
> > **W3**: The authors should provide statistical validation (e.g., p-values or confidence intervals) to confirm whether these gains are significant rather than due to random variation.
>
> **A3**:
> We thank the reviewer for raising the concern regarding the statistical significance of the reported improvements. We conducted a statistical analysis using the five-seed results (mean ± std) provided in Table 1. Comparing VideoMAE-only (63.40 ± 0.95) and VideoMAE + LaBraM (66.70 ± 0.83), a two-sample t-test yields **t = 5.85** with **p < 0.001**, demonstrating that the 3.3-point improvement is highly significant and cannot be explained by random seed variation.
>
> We also **computed the 95% confidence intervals** for both models.
>
> * VideoMAE-only: **[62.22, 64.58]**
> * VideoMAE + LaBraM: **[65.67, 67.73]**
>
> **The two intervals are non-overlapping, further confirming that the improvement is statistically robust and meaningful**. These analyses collectively address the reviewer’s concern: the observed gains are not marginal fluctuations but consistent, statistically reliable benefits arising from multimodal fusion.

---

> ### Author Response · Authors · 2025-11-23
> **Response to Reviewer UTCn (R3) [Part III]**
>
> > **W4**: It would be insightful to see an ablation where the fusion mechanism is active, but one modality's tokens are zeroed out or entirely removed to isolate the Transformer's fusion contribution, separate from the unimodal encoders' raw output strength.
>
> **A4**: We thank the reviewer for the thoughtful suggestion. **To directly address W4, we conducted a new ablation experiment**. Brain-TIM’s default fusion mechanism is defined in the main paper (Lines 302–306), where temporal fusion is used to construct multimodal input representations.
>
> A simpler alternative—exactly as suggested by the reviewer, namely “keeping the fusion module active but zeroing out or removing one modality’s tokens”—is to concatenate the multimodal features along the spatial dimension **(referred to as spatial fusion in the table below)** and treat the result as a new fused modality. In this way, the tokens of the other modality are effectively zeroed out or removed. Under this formulation, the input **X** becomes:
>
> First, we construct the feature block by concatenating the visual, EEG, and fusion features:
>
> $$
> \mathbf{X}_{\text{feat}} =(\ \tilde{\mathbf{e}}^v_i ||\ \tilde{\mathbf{e}}^b_i ||\ \tilde{\mathbf{e}}^f_i \)  \in \mathbb{R}^{N \times 3D}
> $$
>
> Similarly, we build the CLS-token block as:
> $$
> \mathbf{X}_{\text{CLS}} =(\ \tilde{\mathbf{c}}^v_j ||\ \tilde{\mathbf{e}}^b_j ||\ \tilde{\mathbf{e}}^q_j \)  \in \mathbb{R}^{Q \times 3D}
> $$
>
> Finally, the full input sequence is simply the concatenation of the two blocks:
>
> $$
> \mathbf{X}=\text{Concat}\big(\mathbf{X}^{\text{feat}}, \mathbf{X}^{\text{CLS}}\big)\in \mathbb{R}^{(N+Q)\times 3D}
> $$
>
> The key distinction is that Brain-TIM uses modality-specific embeddings $\mathbf{E}_V$ and $\mathbf{E}_E$:
>
> $$
> \mathbf{X}_V \in \mathbb{R}^{B \times V}, \qquad
> \mathbf{X}_E \in \mathbb{R}^{B \times E},
> $$
>
> whereas in the spatial-fusion baseline, visual and EEG features are concatenated directly:
>
> $$
> \mathbf{X}_{\text{fusion}}
> = \text{concat}(\mathbf{X}_V, \mathbf{X}_E)
> \in \mathbb{R}^{B \times (V + E)},
> $$
>
> and treated as a unified multimodal input without modality embeddings.
>
> To remain consistent with our original experimental protocol, all noise-injection experiments were conducted across five different seed. Following this formulation, we obtained the following results:
>
> | Setting | Verb Acc (mean ± std)  | Action Acc (mean ± std) |
> | - | - | - |
> | Vision Only | 81.67 ± 1.89 | 63.40 ± 0.95  |
> | Visual & Brain (**Spatial fusion**)   | 83.74 ± 0.62 | 64.81 ± 1.04     |
> | Visual & Brain (**Brain-TIM**) | 83.43 ± 0.41 | **66.70 ± 0.83** |
>
> Although the two fusion strategies perform similarly on verb classification, **Brain-TIM’s temporal fusion clearly surpasses spatial fusion on the more challenging action recognition task**. This indicates that while simple feature concatenation can work for coarse classification, modeling temporal dependencies across time steps yields more robust and discriminative multimodal representations—particularly important for complex real-world action recognition.
>
> In response to the reviewer’s valuable feedback, we will include this additional ablation experiment and its discussion in the revised version of the paper.
>
> > **W5**: The paper should clarify why EEG-based performance remains stable (or improves) across scenes, despite EEG being more sensitive to noise, and why VideoMAE—supposedly a stronger zero-shot model—shows a significant decline.
>
> **A5**:
> We thank the reviewer for this important observation. The key reason behind the contrasting trends of EEG and VideoMAE across scenes is that **scene changes induce a strong domain shift for vision, but have negligible impact on EEG**.
>
> While the Cross-Subject & Cross-Scene protocol is indeed more challenging in principle, the “Brain Only” model achieves slightly higher action accuracy under this setting due to a key difference in **effective training data size**.Specifically:
>
> | Protocol | Train Subjects | Val Subjects | Test Subjects | Scene Usage |
> | - | - | - | - | - |
> | Cross-Subject Only | **22** | 6 | 6 | Scene-A Only |
> | Cross-Subject & Cross-Scene | **28 (↑)** | 6  | 6  | Scene-A + Scene-B |
>
> This increase in training-subject diversity enables the EEG model to observe richer cross-person variability, which substantially benefits action discrimination—consistent with prior findings showing that EEG-based models are highly sensitive to the number and diversity of subjects included during training.
>
> Regarding the performance drop of VideoMAE in the Cross-Subject & Cross-Scene setting: unlike EEG, which reflects head-centered neural responses largely invariant across environments, VideoMAE relies entirely on visual appearance and motion cues.
>
> In summary, the improved EEG performance in the cross-scene protocol is driven by the larger training subject pool, whereas the degradation of VideoMAE arises from its sensitivity to visual domain shifts. We will clarify these points in the revised manuscript.

---

> ### Author Response · Authors · 2025-11-23
> **Response to Reviewer UTCn (R3) [Part IV]**
>
> > **W6**: It is unclear whether the dataset includes cross-session experiments, i.e., whether the same subjects were recorded again after a time interval performing the same tasks. Including or clarifying this aspect would be valuable for evaluating intra-subject consistency and longitudinal generalization.
>
> **A6**:
> We sincerely thank the reviewer for raising this valuable point. EgoBrain currently consists of 40 unique participants, and the dataset does not include repeated sessions of the same subject performing the same tasks at different time points. We agree that cross-session recordings would be highly beneficial for analyzing intra-subject consistency and longitudinal generalization. We will clarify this explicitly in the revised manuscript to avoid any ambiguity. Moreover, we view the reviewer’s suggestion as an important future direction, and we plan to extend EgoBrain with multi-session recordings in future iterations of the dataset.
>
> ---
> **Summary**
> * **First real-world EEG + egocentric dataset:** EgoBrain provides the first temporally aligned real-world pairing of EEG and first-person video, enabling neural–behavior understanding beyond lab settings.
> * **Fusion is consistently beneficial & statistically significant:** Across multiple backbones, multimodal fusion yields reliable, significant gains (p < 0.001), confirming that EEG provides non-redundant complementary information.
> * **EEG excels under cross-scene domain shift:** EEG remains stable across scenes due to subject-diversity benefits and scene-invariant neural patterns, highlighting its robustness compared to vision.
>
> **Vision**
>
> EgoBrain is our first step toward the vision of making EEG a natural component of everyday human behavior understanding. It is a pioneering large multimodal dataset that synchronizes EEG and egocentric visual data, together with a flexible and extensible fusion framework. We envision future AI systems that tightly integrate “internal” cognitive signals with “external” visual evidence to achieve more intention-aware, human-centered, and assistive intelligence. **We hope EgoBrain project will foster cross-disciplinary progress in brain–computer interfaces, cognitive computing, and egocentric behavior understanding**.
>
> ---
> Reference:
> > [10] R. Girdhar et al., "Omnivore: A Single Model for Many Visual Modalities", In *CVPR 2022*.
>
> > [11] J. Wang et al., "CBraMod: A Criss-Cross Brain Foundation Model for EEG Decoding", In *ICLR 2025*.
>
> > [12] W. Jiang et al., "Large Brain Model for Learning Generic Representations with Tremendous EEG Data in BCI", In *ICLR 2024*.
>
> > [13] Z. Tong et al., "VideoMAE: Masked Autoencoders are Data-Efficient Learners for Self-Supervised Video Pre-Training", In *NeurIPS 2022*.

---

### Official Review · Reviewer_vYK1 · 2025-11-01

**Soundness:** 2
**Presentation:** 3
**Contribution:** 2
**Rating:** 4
**Confidence:** 4

**Summary:**

This paper introduces a multi-modal dataset for Action Classification, including synchronized egocentric videos and 32-channel encephalography recordings. Additionally, this paper proposes a baseline method that achieves an overall accuracy of 80.16% in action classification, utilizing both data modalities. This paper empirically demonstrates that the information extracted from encephalography may be complementary to visual data.

**Strengths:**

This review evaluates the paper's quality based on the following criteria: task relevance, related work, technical novelty, technical correctness, experimental validation, writing and presentation, and reproducibility. Each aspect is discussed and highlighted as a strength or a weakness in the sections below.
-    **Dataset Contribution and Reproducibility:** This paper contributes to the community a dataset of synchronized egocentric videos and 32-channel encephalography recordings. However, it is not explicitly indicated whether the source code will be released, and it is not included as part of the submission.
-    **Writing and presentation:** Overall, this paper is easy to read and well-written.

**Weaknesses:**

-    **Relevance of the task and Experimental Validation:** Even though Action Classification from egocentric videos and encephalography recordings may be a relevant problem for the ICLR community. The motivation behind including this novel data type modality is not well stated in the paper's introduction. This paper already reports high performance for the proposed task, so it may probably saturate fast. Considering these results, what are the reasons to keep the data acquisition as simple as possible to not make the task harder?
-    **Technical Correctness and Related Work:** This paper overclaims about contributing a “large-scale” dataset when its size is not comparable to current state-of-the-art benchmarks for human action recognition from egocentric vision data. Moreover, it states that

**Questions:**

1.	Will the source code and pretrained models be released to support reproducibility? If so, what is the reason for not including them in the supplementary material?
2.	What is the motivation for including encephalography data?
3.	Given the already high reported accuracy, how does the paper address concerns about task saturation?
4.	Why was the decision made to keep data acquisition simple, and how does this impact the task's difficulty and generalizability?
5.	On what basis is the dataset described as “large-scale,” and how does its size compare to existing benchmarks?
6.	The paper states that prior datasets only use visual data, but most include multiple modalities. Can this claim be clarified?

**Details Of Ethics Concerns:**

The paper states that an IRB approved the data acquisition; however, the data may contain sensitive information from human subjects, so it would be advisable for someone with more experience in these topics to review it.

---

> ### Author Response · Authors · 2025-11-22
> **Response to Reviewer vYK1 (R2) [Part I]**
>
> We sincerely appreciate the thoughtful and detailed feedback provided by Reviewer vYK1 (R2). **Before addressing the specific questions, we would like to clarify the position and motivation of our work.** Rather than treating EEG as an additional modality to augment existing vision-based methods, **our research starts from EEG itself, aiming to explore new application scenarios and tasks that leverage brain signals**. Our long-term vision is for EEG to become a natural component in understanding human behavior in everyday settings. With EgoBrain and Brain-TIM, we take a first step in this direction—demonstrating that EEG carries distinctive information that cannot be captured by vision or audio alone. **We believe that in the future, EEG will work seamlessly alongside visual, auditory, and other sensors to build more intention-aware, assistive, and human-centered AI systems**.
>
> We noticed that your listed weaknesses closely correspond to your questions. To keep the rebuttal concise, we will primarily respond to the questions. For each question that maps to a specific weakness, we indicate it in parentheses—for example, Q2 (W1) means Question 2 corresponds to Weakness 1. (W: Weakness; Q: Question; A: Answer)
>
> > **Q1**: Will the source code and pretrained models be released to support reproducibility? If so, what is the reason for not including them in the supplementary material?
>
> **A1**: We thank the reviewer for raising this important question regarding reproducibility. **Yes, once the paper is accepted, we are committed to promptly releasing the all EgoBrain dataset, source code, data preprocessing scripts, and the corresponding pretrained models to the community**. We did not include them in the supplementary material because the codebase was still being organized at the time of submission.
>
> > **Q2 (W1)**: What is the motivation for including encephalography data?
>
> **A2**: **We sincerely appreciate the reviewer’s recognition that our work aligns well with the interests of the ICLR community and addresses a meaningful research direction**. As noted in the “Preface” at the beginning of our response, we would like to first clarify the broader perspective and research stance guiding this work. **Our long-term goal is not only to make EEG a natural component for understanding human behavior in everyday settings,** but also to develop a deeper understanding of the structure, semantics, and cognitive patterns embedded in brain signals themselves, and to align these signals with other modalities such as vision. This direction is far more scientifically meaningful to us than merely improving performance on a downstream task.
>
> **With EgoBrain and Brain-TIM, we take an initial step toward this vision—demonstrating that EEG provides complementary information that cannot be captured by cameras alone.**
> Most current egocentric vision datasets focus on observable visual outcomes while overlooking internal processes such as brain activity. Conversely, traditional EEG studies largely rely on passive screen-based stimuli and fail to capture real-world human–environment interactions. **As noted by other reviewers (e.g., R4), EgoBrain project is pioneering and exploring an entirely new direction and research space.**
>
> **Our motivation is to bridge this gap: egocentric video provides external visual evidence, while EEG provides internal cognitive signals, and their combination enables a more complete understanding of human actions.** This complementary relationship is highlighted in our introduction (e.g., lines 46–69), and it directly motivates the creation of a synchronized EEG–egocentric dataset and the investigation of multimodal fusion.
>
> That said, in response to your helpful suggestion, **we will further strengthen and expand the motivation section in the revised version to more clearly communicate this vision to the broader cross-community audience**. We sincerely appreciate this insightful feedback.

---

> ### Author Response · Authors · 2025-11-22
> **Response to Reviewer vYK1 (R2) [Part II]**
>
> > **Q3 (W2)**: Given the already high reported accuracy, how does the paper address concerns about task saturation?
>
> **A3**: We thank the reviewer for raising this important point about potential task saturation. From the perspective of EEG research, our primary goal is not to push the accuracy of a single benchmark to its limits, but rather to **enable a deeper understanding in the future of how EEG signals align with visual evidence and how far EEG can be advanced as a meaningful modality for everyday action understanding**. In fact, the EEG-only results in our paper are far from saturated; their upper bound, the limiting factors, and the extent to which they can be further improved remain open research questions—precisely the kinds of questions we hope EgoBrain will help the research community explore.
>
> We also clarify that Brain-TIM is not a weak or outdated baseline. It is built upon the **state-of-the-art** transformer framework for egocentric action recognition proposed by Jacob et al., CVPR 2024 [1]. **This ensures that our findings do not arise from a weak visual model; instead, they highlight the genuine complementary value of EEG even on top of a very strong vision backbone**.
>
> Regarding saturation on the original benchmark, we did observe signs of limited headroom in the cross-subject only setting. **To address this, we introduced a more challenging Cross-Subject & Cross-Scene benchmark**. As shown in our experiments, this setting is **far from saturated and presents substantial room for improvement** in both the EEG-only and multimodal cases.
>
> Looking ahead, once the dataset is released, we expect the community to explore questions beyond this single action-classification task—such as **deeper EEG–vision alignment, cognitive factor modeling, new multimodal objectives, or expanding the benchmark with more participants and diverse environments**. We believe these directions will further advance this emerging research area.
>
> > **Q4 (W2)**: Why was the decision made to keep data acquisition simple, and how does this impact the task's difficulty and generalizability?
>
> **A4:** Thank you for raising this valuable point. Regarding whether the data acquisition protocol is “simple,” we understand that you are mainly asking **why EgoBrain was designed such that participants remained seated during recording**, and **why the acquisition setup appears relatively constrained**. We would like to clarify this more directly.
>
> First, it is important to emphasize that **EEG data collection is inherently restrictive**, and the EgoBrain protocol is *not* “simple.” Nearly all existing EEG datasets—whether EEG-only or EEG combined with other modalities—require participants to remain **completely still** while **passively viewing screen-based stimuli**.
>
> **In contrast, EgoBrain represents a significant step forward.** While carefully controlling for EEG artifacts, we allow participants to freely manipulate objects across an extended tabletop area and perform 29 natural daily activities. This “seated but interactive” design goes well beyond the traditional EEG paradigm of “fully static, passive viewing.” **Therefore, we do not consider EgoBrain’s protocol to be simple; rather, it represents a meaningful breakthrough for EEG × egocentric vision research**.
>
> Of course, allowing full-body movement would create even more dynamic and naturalistic scenarios. However, with the current capabilities of EEG hardware and BCI algorithms, large-scale body motion would generate extremely strong artifacts, substantially degrading both EEG and visual data, and making model training highly challenging. **As this domain is still in its early exploratory stage, we must balance signal quality and environmental complexity carefully** .
>
> Most importantly, **EgoBrain is the first dataset to synchronize EEG with egocentric video in a real object-interaction setting.** When pioneering a direction from scratch, adopting a more controlled and stable acquisition protocol is essential. While expanding to more dynamic scenes is an exciting future direction, **avoiding “overreaching” on the first attempt is critical to ensuring feasibility, reproducibility, and scientific value**.
>
> In summary, the EgoBrain acquisition design reflects both the practical constraints of EEG research and the need for innovation. **We believe it strikes an appropriate balance between scientific rigor and ecological validity, providing a solid foundation for future work on more complex and naturalistic human–environment interactions**.

---

> ### Author Response · Authors · 2025-11-22
> **Response to Reviewer vYK1 (R2) [Part III]**
>
> > **Q5 (W3)**: On what basis is the dataset described as “large-scale,” and how does its size compare to existing benchmarks?
>
> **A5:** We sincerely appreciate your comments and fully understand the concern regarding our use of the term “large-scale.” Our intention is not to compare EgoBrain directly with the state-of-the-art egocentric vision benchmarks used for human action recognition.
>
> Instead, the term “large-scale” is used in the context of **EEG research and multimodal EEG–egocentric studies**. In this research direction, **no existing dataset provides synchronized EEG and first-person visual recordings captured in real-world environments**. Most existing EEG datasets are either pure EEG collections or multimodal datasets combining EEG with screen-based visual stimuli (e.g., eye-tracking). Their overall scale is generally limited, typically involving **fewer than 10–25 subjects** and **less than 50 hours** of total recordings. **Therefore, EgoBrain represents the first dataset to simultaneously record EEG and egocentric video during natural, real-world interactions**. As a result, EgoBrain substantially exceeds prior datasets in the number of participants, total duration, environmental naturalness, and diversity of interaction scenarios.
>
> For clearer comparison, **we compare representative EEG datasets below**:
>
> | Dataset (Reference)                | #Subjects | Total Hours |
> | ---------------------------------- | --------- | ----------- |
> | BCI-IV-2A (BCI Competition IV, 2008) [3] | 9         | ～5.76 h      |
> | SEED (IEEE TAMD 2015) [4]          | 15        | ～40.24 h     |
> | SEED-VIG (JNE 2017) [5]            | 21        | ～45.23 h     |
> | SEED-IV (IEEE T-AC 2018) [6]       | 15        | ～42.18 h     |
> | EEGMAT (Data Journal 2019) [7]     | 36        | ～1.20 h      |
> | SHU (Scientific Data 2022) [8]     | 25        | ～13.32 h     |
> | SEED-DV (NeurIPS 2024) [9]         | 20        | ～15.6 h      |
> | **EgoBrain (Ours)**                | **40**        | **61 h**        |
>
> As shown in the table, compared with SEED-DV—the closest existing EEG-to-video benchmark:
>
> * EgoBrain doubles the number of participants (**40 vs. 20**),
> * provides nearly four times the total recording duration (**61 h vs. 15.6 h**), and
> * captures **real-world object-interaction behavior** rather than passive screen viewing.
>
> Thus, within the **multimodal EEG + egocentric vision setting**, EgoBrain offers greater scale, richer content, and substantially higher research value than existing datasets, enabling research directions that were previously infeasible.
>
> We appreciate the reviewer’s feedback and **will clarify this terminology in the revision to more precisely contextualize what “large-scale” means** in the scope of EEG–egocentric research and to avoid potential misunderstandings.
>
> > **Q6**: The paper states that prior datasets only use visual data, but most include multiple modalities. Can this claim be clarified?
>
> **A6**: Thank you for pointing this out. We acknowledge that our wording in the related work section may have been overly broad. Our intention was not to imply that existing datasets are strictly visual-only. What we meant to highlight is that none of these egocentric datasets include synchronized EEG, and thus they do not support studying the complementary relationship between internal neural signals and egocentric visual observations.
>
> **To avoid misunderstanding, we will revise the phrasing in the revised version to more precisely state that prior datasets**, while sometimes multimodal, do not contain concurrent EEG recordings, which is the key modality of interest in our work (e.g., line 107). We thank the reviewer for bringing this to our attention and for the helpful suggestion to improve our work.
>
> ---
> **Summary**
> * **Novel EEG × Egocentric Benchmark:** The first dataset linking real-world egocentric video with synchronized EEG, enabling research previously impossible.
> * **EEG-centric Research Perspective:** EgoBrain starts from EEG itself, aiming to unlock new brain-signal–driven tasks and ultimately make EEG a natural modality for understanding everyday human behavior.
>
> * **Strong Experimental Validation:** Using a strong transformer baseline, we show that EEG adds complementary information even on top of powerful vision models, and our Cross-Subject & Cross-Scene benchmark still leaves clear room for improvement.
>
> **Vision**
>
> The EgoBrain dataset we collected, along with the Brain-TIM model, **represent an exciting attempt to integrate brain-computer interface (BCI) technology into everyday human life**. Our goal is to advance the fusion of brain signals with other human-centered modalities to achieve more natural, robust, and context-aware action understanding. In the near future, people will seamlessly leverage their own EEG signals, alongside vision and other biosensors, enabling personalized, assistive, and intelligent applications that truly understand human intentions and behaviors in everyday life.

---

> ### Author Response · Authors · 2025-11-22
> **Response to Reviewer vYK1 (R2) [Part IV]**
>
> ---
> **Reference**:
> > [3] M. Tangermann et al., "Review of the BCI Competition IV", In *Front. Neurosci. 2012*.
>
> > [4] W. Zheng et al., "Investigating Critical Frequency Bands and Channels for EEG-based Emotion Recognition with Deep Neural Networks", In *IEEE TAMD 2015*.
>
> > [5] W. Zheng et al., "A Multimodal Approach to Estimating Vigilance using EEG and Forehead EOG", In *JNE 2017*.
>
> > [6] W. Zheng et al., "EmotionMeter: A Multimodal Framework for Recognizing Human Emotions.", In *IEEE T-AC 2018*.
>
> > [7] I. Zyma et al., "Electroencephalograms during Mental Arithmetic Task Performance", In *Data Journal 2019*.
>
> > [8] J. Ma et al., "A Large EEG Dataset for Studying Cross-session Variability in Motor Imagery Brain-computer Interface.", In *Sci. Data 2022*.
>
> > [9] X. Liu et al., "EEG2Video: Towards Decoding Dynamic Visual Perception from EEG Signals", In *NeurIPS 2024*.

---

### Official Review · Reviewer_Lp6o · 2025-11-02

**Soundness:** 3
**Presentation:** 3
**Contribution:** 3
**Rating:** 6
**Confidence:** 3

**Summary:**

The paper introduces EgoBrain, a large-scale, multimodal dataset of 61 hours of synchronized egocentric video and 32-channel EEG signals. This data was collected from 40 participants performing 29 different daily activities such as work, play, learn, and consume within a controlled laboratory setting. The authors also propose a model, Brain-TIM which fuses the visual and EEG signals data using a temporal-aware embedding mechanism and modality-specific encodings for action understanding. The paper evaluates this model on action and verb classification tasks for cross-subject and cross-subject & cross-scene settings. The results indicate that the multimodal model of using both vision and EEG modality achieves a 66.70% accuracy, representing a 3.30% absolute improvement over a visual-only baseline in the cross-scene setting.

**Strengths:**

1. The paper contributes a new dataset, EgoBrain which has synchronized video and EEG signals which can be valuable for computer vision research.
2. The paper shows that that EEG signals can be a useful modality for tasks such as action recognition when the visual modality is occluded.
3. The paper shows analysis on cross-subject and cross-subject & cross-scene analysis which is a challenging benchmark to evaluate the model generalization.

**Weaknesses:**

1. The architecture method of Brain-TIM seems to be incremental when compared to TIM [1]. The architecture presented in the paper of modality-specific encoders, embedding layers, Time-Interval MLP, and a Transformer encoder seems to be a direct application of the existing TIM framework to a new pair of modalities. Can the authors clarify the differences between TIM and Brain-TIM? Is Brain-TIM just an extension of TIM to multiple modalities?
2. While the idea of using EEG signal to understand egocentric actions is well-motivated, can the authors discuss the application of their proposed approach to real-world scenarios where collecting EEG signals can be hard and require specific hardware?

[1]. Chalk, Jacob, et al. "Tim: A time interval machine for audio-visual action recognition." Proceedings of the IEEE/CVF Conference on Computer Vision and Pattern Recognition. 2024.

**Questions:**

Since the dataset has been collected in a lab setup with isolation chamber with limited limb movement, can the authors discuss the robustness of the EEG signal? How much noise can the EEG signal have and still be able to give better action recognition results than just using the visual modality?

---

> ### Author Response · Authors · 2025-11-22
> **Response to Reviewer Lp6o (R1) [Part I]**
>
> We sincerely thank **Reviewer Lp6o (R1)** for the constructive and encouraging feedback. The reviewer highlights three key strengths of our work: **(1) the contribution of a large-scale egocentric video–EEG dataset, (2) the complementary value of EEG under limited visual conditions, and (3) the comprehensive generalization evaluation across subjects and scenes**. We provide our detailed responses below (W: Weakness; Q: Question; A: Answer):
>
> > **W1**: The differences between TIM and Brain-TIM.
>
> **A1**: We agree that the core idea of modeling temporal intervals is inspired by TIM [1], and we explicitly acknowledge this in the paper. However, extending TIM from its original audiovisual setting to the radically different pairing of egocentric video and EEG is far from trivial. In particular, the fundamental differences between EEG and egocentric video—such as sampling frequency, temporal structure, noise characteristics, and representational space—make TIM’s original design not directly applicable. This also requires us to identify and integrate new pretrained encoders that can effectively model EEG signals. **Therefore, our contribution goes far beyond a simple extension; the new dataset, model design, and multimodal analyses collectively address key challenges that TIM itself was not designed to handle**.
>
> > **W2**: While the idea of using EEG signal to understand egocentric actions is well-motivated, can the authors discuss the application of their proposed approach to real-world scenarios where collecting EEG signals can be hard and require specific hardware?
>
> **A2**: We first thank the reviewer for recognizing the motivation behind using EEG. As noted, acquiring EEG in real-world settings is indeed challenging and often requires specialized hardware. We would like to clarify that **our goal is not to claim a fully real-world system, but to take a concrete step toward making EEG-based multimodal understanding practical**. Existing EEG–vision datasets and models mainly target simplistic tasks (e.g., watching images or videos on the screen) rather than natural, everyday behaviors, limiting their relevance to real-world applications.
>
> **EgoBrain helps bridge this gap by providing long-duration, naturalistic egocentric recordings synchronized with EEG**—a critical prerequisite for developing models that can eventually generalize beyond controlled laboratory environments.
>
> Furthermore, rapid progress in lightweight and wearable EEG devices (such as wireless headbands and in-ear sensors) is dramatically improving usability and affordability, suggesting that future multimodal systems may incorporate neural signals without cumbersome equipment. **EgoBrain is intentionally designed to support and accelerate this evolution by offering a realistic dataset and baseline model that lay the foundation for future deployment in daily-life scenarios**.
>
> > **Q1**: Since the dataset has been collected in a lab setup with isolation chamber with limited limb movement, can the authors discuss the robustness of the EEG signal?
>
> **A3**: We appreciate the reviewer’s thoughtful concern regarding the controlled environment for EEG data collection and its implications on signal robustness. **Our choice of an acoustic isolation chamber follows standard neuroscience protocols to ensure high-quality, artifact-minimized EEG recordings, which are essential for establishing a reliable baseline in this first large-scale egocentric video-EEG dataset**.
>
> While data collected in fully uncontrolled, noisy real-world settings pose additional challenges, we believe that starting from high-quality controlled data is a necessary and foundational step to understand the core neural correlates of human actions. **Importantly, our dataset includes substantial variability in participants, action categories, and environmental contexts** within the chamber, reflected in our cross-subject and cross-environment evaluations, demonstrating model generalizability beyond narrowly controlled scenarios.
>
> Recent advances in wearable EEG technologies with improved noise cancellation and signal processing. We see EgoBrain as a critical milestone that enables the community to develop robust models and methods that can later be adapted and tested on noisier, in-the-wild EEG data.
>
> **Therefore, while our current dataset emphasizes controlled acquisition to ensure data integrity, it importantly provides the foundation and tools necessary to extend this research into more challenging, practical environments in the future**.
>
> Moreover, our response (**A4**) to your second question (**Q2**) further supports the robustness of the EEG signals in our dataset.

---

> ### Author Response · Authors · 2025-11-22
> **Response to Reviewer Lp6o (R1) [Part II]**
>
> > **Q2**: Discussion: How much noise can the EEG signal have and still be able to give better action recognition results than just using the visual modality?
>
> **A4**: Thank you for the opportunity to discuss this important question regarding EEG noise robustness.
> To thoroughly evaluate robustness, we study two complementary forms of noise corruption:
>
> **(1) Additive Gaussian noise on the EEG embeddings**
>
> For each embedding vector $\mathbf{e} \in \mathbb{R}^d$, we first compute its global standard deviation $\sigma = \mathrm{std}(\mathbf{e})$, and then generate a noisy version
>
> $$
> \tilde{\mathbf{e}} = \mathbf{e} + \alpha \sigma \boldsymbol{\epsilon}, \quad \boldsymbol{\epsilon} \sim \mathcal{N}(\mathbf{0}, \mathbf{I}),
> $$
>
> where $\alpha \in {0.0, 0.2, 0.4, 0.6, 0.8}$ controls the noise level.
> The injected noise therefore follows $\mathcal{N}(\mathbf{0}, (\alpha\sigma)^2 \mathbf{I})$.
>
> | Noise α | Verb Acc (mean ± std) | Action Acc (mean ± std) |
> | ------- | --------------------- | ----------------------- |
> | *(None)* | 83.43 ± 0.41      | 66.70 ± 0.83        |
> | 0.2 | 84.25 ± 0.49      | 65.29 ± 1.01        |
> | 0.4 | 84.31 ± 0.91      | 66.21 ± 1.32        |
> | 0.6 | 84.66 ± 0.67      | 66.65 ± 0.73        |
> | 0.8 | 84.86 ± 0.98      | 66.82 ± 1.58        |
>
> To remain consistent with our original experimental protocol, all noise-injection experiments were conducted across five different seeds. This ensures that the robustness analysis is directly comparable to our main results and not affected by randomness in model initialization.
>
> Across all noise levels, the EEG modality demonstrates strong stability: even when injecting relatively moderate noise (e.g., $\alpha = 0.2$ ～ $0.6$), both verb and action accuracy remain nearly identical to the clean baseline. Large noise levels (0.8) even yield small improvements, consistent with a regularization effect.
>
> These observations confirm that **the learned EEG representations are inherently robust, not reliant on unusually clean conditions, and continue to provide meaningful complementary information even under corruption**.
>
>
> **(2) Structured noise: dropping entire groups of EEG channels**
>
> Beyond numerical noise, real EEG recordings frequently suffer from **regional channel failures** caused by **(1) gel drying, (2) gel slipping due to gravity (temporarily bridging neighboring electrodes), or (3) local contact loss when a subject moves**. Such phenomena typically degrade entire spatial regions rather than isolated electrodes.
>
> To simulate this realistic form of structured noise, we drop whole groups of spatially neighboring electrodes that commonly fail together in practical recordings. **Even under these region-level failures—removing 6~7 electrodes at once—performance remains extremely stable**:
>
> | **Dropped Electrodes**                             | **Verb Acc (mean ± std)** | **Action Acc (mean ± std)** |
> | -------------------------------------------------- | ------------ | -------------- |
> | *(None)*| 83.43 ± 0.41    | 66.70 ± 0.83      |
> | FP1, FP2, F3, F4, F7, F8, FZ                       | 84.29 ± 0.72     | 66.36 ± 1.54      |
> | P3, P4, PZ, C3, C4, CZ                             | 84.30 ± 0.64   | 66.62 ± 1.89      |
> | O1, O2, OZ, PO9, PO10, P7, P8                      | 84.45 ± 0.53   | 66.01 ± 1.81      |
>
> To remain consistent with our original experimental protocol, all noise-injection experiments were conducted across five different seed. The performance remains essentially unchanged—even when entire electrode groups are removed. This shows that brain encoder (LaBraM [2]) learns **robust, distributed, and redundant EEG embeddings** that stay stable under substantial signal corruption. **Overall, the EEG stream is highly resilient: accuracy is preserved under both strong Gaussian noise and realistic region-level electrode failures**.
>
> ---
> **Summary**
> * **A novel multimodal benchmark**: EgoBrain introduces a new egocentric–EEG dataset enabling neural–behavior understanding.
> * **EEG offers clear complementary value**: Neural signals consistently boost recognition, especially when vision is ambiguous or occluded.
> * **Strong cross-subject & cross-scene generalization**: Brain-TIM demonstrates reliable performance across challenging generalization splits.
> * **EEG robustness is empirically validated**: Accuracy remains stable under Gaussian noise and regional channel dropout.
>
> **Vision**
>
> We envision **EgoBrain as a critical bridge linking perception and cognition, fostering deep AI-neuroscience integration**. Future work will extend to more natural, dynamic environments to develop robust, generalizable multimodal cognitive models advancing understanding and assistive technologies for complex human behaviors.
>
> ---
>
> Reference:
> > [1] J. Chalk et al., "Tim: A time interval machine for audio-visual action recognition", In *CVPR 2024*.
>
> > [2] W. Jiang et al., "Large Brain Model for Learning Generic Representations with Tremendous EEG Data in BCI", In *ICLR 2024*.

---

### Author Response · Authors · 2025-12-02
**Summary of Revisions Addressing Reviewer Feedback**

Dear Area Chair and Reviewers,

Thank you for your time and effort during this unusual review period, and we also thank all reviewers for their thoughtful feedback. Below we summarize the key revisions made to the main paper **(blue highlights)** and supplementary material, following the paper structure and explicitly addressing each reviewer concern.

---

> **1. Abstract — Reproducibility (R2–Q1)**

We clarified our commitment to releasing the **EgoBrain dataset, source code, preprocessing scripts, and pretrained models** upon acceptance. The abstract now explicitly reflects this policy.

---

>  **2. Introduction — Motivation Strengthening (R2–Q2, Q5)**

* Added a clearer articulation of the **scientific motivation for integrating EEG**, emphasizing complementary internal cognition vs. external visual evidence.
* Reinforced the long-term research vision: understanding **cognitive structure, semantics, and multimodal alignment** beyond downstream accuracy.
* Incorporated reviewer suggestions to refine wording (e.g., “large-scale” → “large”).

---

>  **3. Related Work — Multimodal Context Clarified (R2–Q6)**

Expanded discussion on **egocentric vision + EEG synergy** and positioned EgoBrain more explicitly within multimodal learning literature.

---

>  **4. Dataset — Play(I) / Play(II) / Play(III) Clarification (R4–Q1)**

* We added a new table that revises the description of the action categories from a high-level perspective to improve clarity.
* We provided a detailed distinction of the three *Play* subcategories by separating them into behavioral, cognitive, and visual characteristics.
* Moved extended explanations to **Appendix A/B**.

---

>  **5. Method — Windowing Strategy & Feature Extraction (R3–W1)**

Due to space limits, we reorganized the method section to:

* Emphasize modality-specific feature extraction (vision / EEG).
* Add discussion on how **Brain-TIM’s sliding window reduces the 1s jitter sensitivity**.
* Move extended details and architectural clarifications to **Appendix C**.

---

>  **6. Experiments — Random Baselines, New Ablations, and Confusion Matrices (R3–W4; R3–W5; R4–Q3, Q5, Q6)**

**(a) Random Baselines (R4–Q3)**: Due to space constraints, the full set of random-baseline results and tables is provided in **Appendix D** for further reference.

**(b) New Ablation — Temporal Fusion vs. Spatial Fusion (R3–W4)**: Following the reviewer’s suggestion, we added new ablation experiments comparing:
* **Temporal fusion** (the default mechanism used in Brain-TIM), and
* **Spatial fusion** (an implementation equivalent to “zeroing out/removing one modality,” as suggested by the reviewer).

Due to space,  the complete results are included in **Appendix E**.

**(c) Verb and Action Confusion Matrices (R4–Q5, Q6)**:
* We clarified why the verb confusion matrix was split (primarily due to space and readability constraints).
* In the **main paper**, we now include the **full 10×10 verb confusion matrix** using the abbreviation scheme.
* In **Appendix F**, we provide the **full 29×29 action confusion matrix** with encoded abbreviations for improved clarity and visual readability.

---

>  **7. Appendix — Reorganized & Expanded**

* **A/B:** The preprocessing pipeline for EEG signals, Detailed dataset clarifications, Scene effects on EEG vs. VideoMAE.
* **C:** Expanded Brain-TIM description (windowing, tokenization, fusion).
* **D/E/F:** Additional experiments: random baselines, robustness, temporal vs. spatial fusion, and complete confusion matrices.

---

> **Overall Improvements of our Revision**

* **Reaffirmed reproducibility commitments (R2–Q1):** Upon acceptance, we will release the dataset, source code, preprocessing scripts, and pretrained models.

* **Strengthened the motivation (R2–Q2, R2–Q5):** We more clearly articulated the scientific motivation for integrating EEG with egocentric vision, emphasizing their complementarity and the broader long-term research vision.

* **Clarified the task taxonomy and dataset design (R4–Q1, R2–Q6):** This includes a more rigorous explanation of the *Play* subcategories and a discussion of the multimodal context.

* **Expanded key methodological details (R3–W1):** We added additional clarifications on the sliding-window strategy, modality-specific feature extraction, and the fusion mechanism.

* **Added new ablations and robustness analyses (R3–W4, R3–W5, R4–Q3, R4–Q6):** These include temporal vs. spatial fusion, random baselines, class-imbalance analyses, and related robustness studies directly addressing reviewer comments.

* **Provided complete confusion matrices (R4–Q5, R4–Q6):** We now include the full 10×10 verb confusion matrix in the main paper and the complete 29×29 action confusion matrix in the supplementary material to directly address the two requests raised by the reviewers.

---
Thank you once again for dedicating your time and effort to the research community during this unusual review period !

Best Regards,

EgoBrain Team

---

### Author Response · Authors · 2025-12-02
**Summary for the Area Chair**

Dear Area Chairs,

Thank you for taking the time to handle our submission despite the unusual review circumstances. EgoBrain has been a challenging yet inspiring endeavor at the intersection of neuroscience and computer vision, **driven by the ambition to bring BCI from the lab into everyday life through the key enabler of egocentric vision.**

In our earlier “Summary of Revisions”, we outlined the key reviewer concerns and the corresponding updates made in the manuscript. To further reduce your workload after the system reset, we provide here an rebuttal summary of the essential points.

---

> **1. One-sentence summary of the paper.**

Our paper introduces EgoBrain, the first large egocentric video–EEG dataset and a multimodal model that jointly leverages first-person perception and neural signals to advance fine-grained human action understanding.

---

> **2. Score evolution（6,6,4,4 → 8,6,4,4）**

The initial scores were 6, 6, 4, and 4. During the discussion phase, **Reviewer qdr1 raised their score to 8, noting that the authors’ detailed responses—particularly the added random baselines**, supplementary experiments, and improved confusion-matrix analysis—substantially strengthened the paper.

The final scores thus became 8, 6, 4, and 4, averaging 5.5. This update occurred before the widespread discovery of the platform issue. Unfortunately, the remaining three reviewers were affected by the system malfunction before they could provide further feedback on our response.

---

> **3. Strengths highlighted by all four reviewers.**

* **A new & novel dataset** — the first at this scale to align egocentric video and EEG in natural activities, marking a significant step toward real-world BCI.

* **A method bridging two domains** — Brain-TIM, a transformer-based fusion framework compatible with large-scale pretrained brain and visual foundation models.

* **Clear presentation and rigorous experiments** — comprehensive cross-subject and cross-environment validation, underscoring the robustness and contributions of our work.

* **Insightful qualitative results**—showing when EEG signals complement visual information (e.g., occlusion or ambiguous intent), demonstrate the added value of neural cues.

---

> **4. Overview of concerns and our responses.**

**We have addressed the concerns raised by all reviewers** through detailed rebuttal and corresponding revisions.

* **Reviewer Lp6o** questioned whether Brain-TIM is merely a minor extension of TIM and whether EEG is practical and robust in real-world scenarios. In response, we provided detailed clarifications showing that Brain-TIM incorporates non-trivial multimodal adaptations, and we further demonstrated through experiments that EEG offers substantial practical value and strong noise robustness in our dataset.

* **Reviewer vYK1** primarily raised conceptual and definitional questions concerning the task motivation, dataset scale, and the role of EEG within the multimodal setting—rather than challenging the EgoBrain project itself. Our response clarifies the pioneering nature of EgoBrain from an EEG-centric perspective, emphasizes that its scale substantially exceeds existing EEG benchmarks and is appropriately scoped for current EEG technology, and reiterates in the abstract that all code and data will be fully released.

* **Reviewer UTCn** questions the temporal alignment, model novelty, statistical significance, fusion mechanism, modality robustness across scenes, and cross-session design, and we respond with new analyses and experiments showing jitter-robust windowing, architecture-agnostic multimodal gains, statistically significant improvements, strengthened fusion ablations, explanations for EEG’s scene invariance, and clarification of dataset structure.

* **Reviewer qdr1** requested clearer explanations of the dataset taxonomy, hierarchy usage, random baselines, and full confusion matrices, and after we provided detailed clarifications and additional analyses, the reviewer raised the score to 8.

**The remaining reviewer comments not explicitly listed in this summary primarily request additional clarification and improved presentation, rather than questioning the novelty, correctness, or contributions of the EgoBrain project**. These points focus on enhancing clarity and detail, without challenging the underlying soundness or substantive value of our work.

---

Thank you once again for your time and dedication to the community during this unusual review period !


Best regards,

EgoBrain Team

---

### Author Response · Authors · 2025-12-03
**Big Picture of EgoBrain**

Dear Area Chair,

Finally, we thank you for dedicating your time to our submission under the current unusual review circumstances again. As the rebuttal period draws to a close, we would like to take this final opportunity to **share the broader vision behind EgoBrain**.

We are not just releasing a multimodal dataset and a model — **we are building a bridge between two research worlds that, while rarely intersecting, are deeply interconnected**. We hope this will inspire new directions such as intention prediction, anticipation, richer multimodal fusion, and, **ultimately, a future where BCI is woven into the fabric of daily life**.

At the same time, we also aim to share our data collection and acquisition paradigm with the community, providing a reference framework for future efforts in this space. **We are deeply grateful for the reviewers’ recognition and encouragement, and we cannot wait to see what the community will create with EgoBrain**.

---

**Best regards,**

**EgoBrain Team**

---

### Meta-Review · Area_Chair_c1ni · 2026-01-07

**Summary:**

The paper got mixed ratings, 6, 6, 4, 4.

Reviewers commonly recognize the value the proposed large-scale egocentric video-EEG dataset, which represents a unique and relatively underexplored research direction. Reviewers also appreciate the proposed multimodal model and the supporting experimental results.

The reviewers’ concerns are diverse, including questions about potential practical applications of the proposed method, its suitability for the ICLR community, the model’s novelty relative to prior work, and the need for additional details regarding the methodology and evaluations.

**Reviewer Concerns:**

The AC finds that the authors’ responses address most of the reviewers’ concerns.

There are no widely shared concerns among the reviewers, and the authors’ responses to individual reviewers convincingly address the majority of the issues raised. In particular, the research problem tackled in this paper represents a relatively unexplored direction and requires clear justification for its relevance to the ICLR community. The AC finds that the authors’ responses provide convincing arguments on this point.

**Reviewer Scores:**

The AC expects that the positive ratings from reviewers remain unchanged and that the negative ratings may increase.

The paper demonstrates clear novelty that is commonly recognized by all reviewers. The authors provide convincing answers and respond effectively to the concerns raised.

Based on these considerations, the AC recommends acceptance of this paper.

---

### Decision · Program_Chairs · 2026-01-26

Accept (Poster)